# ClimateGAN: Raising Climate Change Awareness by Generating Images of Floods

Victor Schmidt[*1,2], Alexandra Luccioni[*1,2], Mélisande Teng[1,2], Tianyu Zhang[1,2],
Alexia Reynaud[1,2], Sunand Raghupathi[1,3], Gautier Cosne[1,2], Adrien Juraver[1,2], Vahe Vardanyan[4],
Alex Hernández-García[1,2], and Yoshua Bengio[1,2]

[1]Mila Québec AI Institute, Montréal, Canada
[2]Université de Montréal, Montréal, Canada
[3]Columbia University, New York City, USA
[4]CDRIN, Matane, Canada

## Abstract

Climate change is a major threat to humanity and the actions required to prevent its catastrophic consequences include changes in both policy-making and individual behaviour. However, taking action requires understanding its seemingly abstract and distant consequences. Projecting the potential impacts of extreme climate events such as flooding in familiar places can help make the impacts of climate change more concrete and encourage action. As part of a larger initiative to build a website (`https://thisclimatedoesnotexist.com`) that projects extreme climate events onto user-chosen photos, we present our solution to simulate photo-realistic floods on authentic images. To address this complex task in the absence of suitable data, we propose ClimateGAN, a model that leverages both simulated and real data through unsupervised domain adaptation and conditional image generation. In this paper, we describe the details of our framework, thoroughly evaluate the main components of our architecture and demonstrate that our model is capable of robustly generating photo-realistic flooding on street images.

## 1 Introduction

Climate change is a serious danger to our societies, with warming temperatures causing extreme weather events that affect the livelihood of an increasing number of people globally (Hoegh-Guldberg et al., 2018). In particular, rising sea levels, increasing precipitation and faster snow melt exacerbate extreme floods, presenting a major risk to  populations worldwide (Dottori et al., 2016). One common barrier to climate action and behavioral change is *distancing*, a psychological phenomenon resulting in climate change being perceived as temporally and spatially distant and uncertain. Showing or simulating first-person perspectives of climate change-related extreme weather events can contribute to reducing distancing (Chapman et al., 2016; Sevillano et al., 2007) and information technologies are increasingly used for this purpose (Herring et al., 2017; Ahn et al., 2014), but they often target specific regions or render manually their effects.

In this context, we have developed *ClimateGAN*, which can generate extreme flooding based on arbitrary street-level scenes, such as Google Street View images. We generate floods of 1 m, a realistic expected water level for climate change-related flooding events (Kulp & Strauss, 2019), and we divide the task of flooding into two parts: a *Masker* model to predict which pixel locations of a given image would be under water if a flood occurred, and a *Painter* model to generate contextualized water textures conditioned on both the input and the Masker's prediction. Our contributions are: proposing and motivating the novel task of street-level flood generation, a data set of pairs of images with/without flooding from a virtual world, the ClimateGAN model which includes a novel multi-task architecture for generating geometry- and semantics-informed binary masks and a procedure to thoroughly evaluate it in the absence of ground-truth data. We also compare our model to existing generative modeling frameworks and provide an ablation study of the components of our model.

---

[*]Corresponding authors - [`schmidtv, luccionis`]`@mila.quebec`

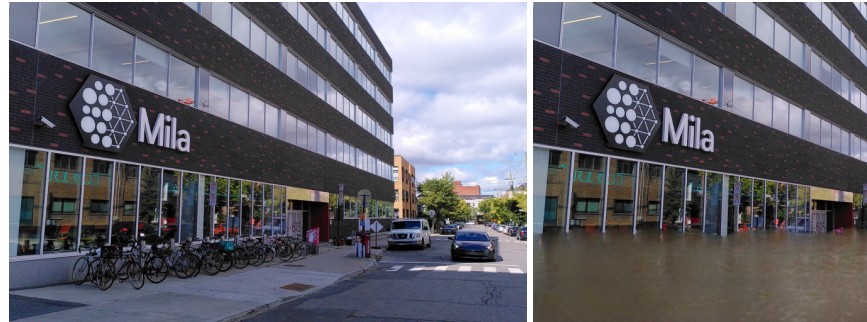

Figure 1: *ClimateGAN, a model that generates extreme floods (right) on street-level images (left).*

## 2 RELATED WORK

While the task of generating extreme flooding in street-level images is novel, related work has been carried out in applying Deep Learning for flood segmentation (Sazara et al., 2019) and flood depth estimation (Kharazi & Behzadan, 2021). Generative modeling has also been used for transferring weather conditions on street scenes using both imagery (Li et al., 2021) and semantic maps (Wenzel et al., 2018). For the purposes of our task and given its unique constraints, we frame our approach in the context of *image-to-image translation*, involving *conditional image synthesis* and *domain adaptation*. We present relevant related work from these three areas in the paragraphs below.

**Image-to-image translation** (IIT) is a computer vision task whose goal is to map a given image from one domain to another (Liu et al., 2017). IIT approaches can either carry out the translation on the entire input image or utilize masks to guide the translation task. In the first case, initial IIT approaches relied on the existence of two aligned domains such as photographs and sketches of the same objects. CycleGAN (Zhu et al., 2017) relaxed this constraint, allowing the domains to remain unaligned, and further progress was made by architectures such as MUNIT (Huang et al., 2018) and CUT (Park et al., 2020). The second category of IIT focuses the translation process on particular input image areas, typically by leveraging attention or segmentation masks. This more closely resembles our case, as we aim to flood only part of the image. Notable examples of this category include Attention-Guided GANs (Tang et al., 2019) and InstaGAN (Mo et al., 2019) which uses instance-level semantic masks to guide the translation process.

**Conditional image synthesis** differs from IIT in that the input can be a label, text or a segmentation map, instead of another image (Mirza & Osindero, 2014). One approach from this category that is particularly relevant to our work is SPADE (Park et al., 2019), a module that enables the transformation of a semantic layout—such as that of a street scene or landscape—into an image that semantically matches this layout. The idea behind SPADE is to create residual blocks where the input is first normalized and then denormalized in a spatially relevant way by small convolutional networks, functions of spatial conditioning variables. This approach also introduced the GauGAN (Park et al., 2019), generator, which leverages SPADE blocks to learn a spatially-adaptive transformation, enabling the synthesis of realistic images based on the input maps.

**Domain adaptation** aims at transferring knowledge from one domain to another using different data sources (Ganin & Lempitsky, 2015). This can be particularly useful in tasks where more (labeled) data is available from a simulated world than in the real world, like in our case. Domain adaptation techniques can then be used to bridge the distributional gap between real and simulated scenes, learning useful tasks such as semantic segmentation and depth prediction, which function both in real and simulated scenarios. Examples of domain adaptation approaches that adopt these techniques include: CYCADA (Hoffman et al., 2018), which leverages cycle-consistency constraints similar to those proposed by CycleGAN to improve domain adaptation, ADVENT (Vu et al., 2019a), which uses Adversarial Entropy Minimization to achieve high performance in unsupervised domain adaptation for semantic segmentation, and Depth-aware Domain Adaptation (DADA) (Vu et al., 2019b), which improves on ADVENT by leveraging dense depth maps.

## 3 CREATING IMAGES OF FLOODS

Our task resembles that of unsupervised image-to-image translation. However, we identified three major challenging differences: first, the translation is restricted to the portion of the image that would contain water rather than altering the image globally. Second, the water occludes multiple objects in the scene and typically only a part of them, which differs from the application cases of instance-aware methods. Finally, we are only concerned with adding water and not the reverse, which eliminates the need for cycle-consistent approaches. Therefore, in order to undertake this task, we developed a novel conditional image synthesis method that consists of two models: a Masker that produces a binary mask of where water would plausibly go in the case of a flood, and a Painter that renders realistic water given a mask and an image. We provide an overview of this procedure in Fig. 2 and describe the individual components in the remainder of this section.

### 3.1 DATA

First-person images of floods are scarce, the corresponding image before the flood is rarely available and even more so scene geometry and semantic segmentation annotations, which we want to leverage during training. To overcome these limitations, we created a virtual world to generate annotated flooded and non-flooded pairs of images, and pursued multiple approaches to collect real photos.

**Simulated Data**   We created a 1.5 $km^2$ virtual world using the Unity3D engine. To be as realistic as possible, we simulated urban, suburban and rural areas, which we flooded with 1m of water to gather 'with' and 'without' pairs (see Appendix A). For each pair of images, we also captured the corresponding depth map and semantic segmentation layout of the scene. Overall, we gathered approximately 20,000 images from 2,000 different viewpoints in the simulated world, which we used to train the Masker. We make this data set publicly available[1] to enable further research.

**Real Data**   Approaches for gathering real flooded street-level images spanned from web-scraping to crowd-sourcing via a website and a mobile app [2] . We also included images without floods of typical streets and houses, aiming to cover a broad scope of geographical regions and types of scenery: urban, suburban and rural, with an emphasis on images from the Cityscapes Cordts et al. (2016) and Mapillary Neuhold et al. (2017) data sets. We collected a total of 6740 images: 5540 non-flooded scenes to train the Masker, and 1200 flooded images to train the Painter.

### 3.2 MASKER

To leverage the information available in the simulated domain and transfer performance to real data, we adapted DADA (Vu et al., 2019b) to train the Masker. The main objective of this procedure is to inform segmentation with depth. A naive use of DADA for our task would miss a crucial difference: the Masker produces information about what *could be* in an image, not what *is present* in it. The core contribution of our Masker's architecture is therefore to structure it as a multi-headed network with depth and segmentation decoders to learn what *is present* in the current scene, then conditioning the flood mask decoder on this information (and the input image) to predict where water *could be*.

In the following, subscripts $s$ and $r$ identify the simulated and real domains, respectively; we use $i \in \{r, s\}$ to refer to an arbitrary domain. $E$ is an encoder network while $D$, $S$ and $M$, are the depth, segmentation and flood mask decoders, respectively, as per Fig. 2.

**Depth decoder**   We consider depth in the disparity space, and predict the normalized inverse depth $\boldsymbol{d}_i = D(E(\boldsymbol{x}_i))$ from an input image $\boldsymbol{x}_i$. We used the scale-invariant loss from MiDaS (Lasinger et al., 2019), which is composed of a scale and shift-invariant MSE loss term $\mathcal{L}_{SSIMSE}$ and a gradient matching term $\mathcal{L}_{GM}$. We used the following targets to compute this loss: ground-truth depth maps for simulated input images and pseudo labels inferred from the MiDaS v2.1 model for real input images. The complete depth loss is:

$$\mathcal{L}_{Depth} = \lambda_1 \mathcal{L}_{SSIMSE} + \lambda_2 \mathcal{L}_{GM}. \tag{1}$$

---

[1]`https://github.com/cc-ai/mila-simulated-floods`
[2]`https://climatepix.mila.quebec/`

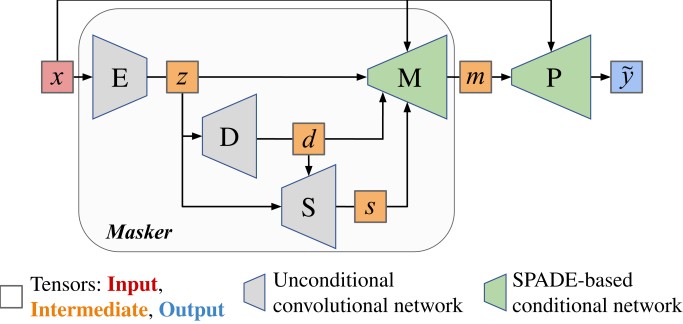

Figure 2: The ClimateGAN generation process: first, the input $x$ goes through the shared encoder $E$. Three decoders use the resulting representation $z$: $D$ predicts a depth map $d$, $S$ produces a depth-informed segmentation map $s$, and lastly $M$ outputs a binary flood mask, taking $z$ as input and sequentially denormalizing it with SPADE blocks conditionally on $d$, $s$ and $x$. Finally, the Painter $P$ generates an image $\tilde{y}$ of a flood, conditioned on the input image $x$ and the binary predicted mask $m$. Note that the Masker and the Painter are trained independently and only combined at test-time.

**Segmentation decoder**  The segmentation decoder $S$ is implemented such that $S \circ E$ corresponds to the DeepLabv3+ architecture (Chen et al., 2018). It is trained as described in DADA, leveraging depth information available in the simulated world to improve segmentation predictions by giving more attention to closer objects, producing $s_i = S(E(x_i), d_i)$. Two fusion mechanisms encourage this: *feature fusion*, which multiplies element-wise the latent vector $z_i = E(x_i)$ by a depth vector obtained from the depth decoder, and *DADA fusion*, which multiplies the *self-information map* (Vu et al., 2019a) $I(s_i) = -s_i \cdot \log s_i$ element-wise with the depth predictions $d_i$ to obtain the depth-aware self-information map $\hat{I}(s_i) = I(s_i) \odot d_i$.

In addition to DADA, we used pseudo labels inferred from a pre-trained segmentation model in the real domain. Otherwise the training of $S$ is similar to DADA: to encourage confident real domain predictions and reduce the gap with simulated predictions an entropy minimization (EM) term $\mathcal{L}_{EM}(s_r)$ is added. Further, WGAN-based adversarial training (Arjovsky et al., 2017) is leveraged to shrink the domain gap between the distributions of real and simulated self-information maps:

$$\mathcal{L}_{Seg} = \lambda_3 \mathcal{L}_{CE} + \lambda_4 \mathcal{L}_{EM} + \lambda_5 \mathcal{L}_{WGAN}. \tag{2}$$

**Flood mask decoder**  This decoder is structured to be conditioned not only on the input image, but also on predictions $d_i$ and $s_i$ from other decoders. To implement this dependence, we propose a new use of SPADE conditional blocks. In our case, for an input $x_i$, the conditioning variable is therefore $U_i = [x_i, d_i, s_i]$, where the tensors are concatenated along the channel axis. The mask $m_i = M(z_i, U_i)$ and its self-information map $I(m_i)$ are computed from the latent representation $z_i = E(x_i)$. We also implemented a total variation (TV) loss on the mask $m_i$ for both domains in order to encourage the predictions to be smooth, ensuring that neighboring pixels have similar values (Johnson et al., 2016)—note that $\Delta$ is the spatial difference of the image mesh:

$$\mathcal{L}_{TV}(m_i) = \mathbb{E}_{n,h,w}[(\Delta_h m_i)^2 + (\Delta_w m_i)^2]. \tag{3}$$

In the simulated domain, we used a binary cross-entropy loss $\mathcal{L}_{BCE}(y_{m_s}, m_s)$ with the ground-truth mask $y_{m_s}$. In the real domain, absent of any ground truth, we encouraged the predicted flood mask $m_r$ to at least encompass the ground by introducing a ground intersection (GI) loss, penalizing masks that assign a low probability to locations where a pre-trained model detected ground $g_r$:

$$\mathcal{L}_{GI}(g_r, m_r) = \mathbb{E}_{n,h,w}[\mathbb{1}_{(g_r - m_r) > 0.5}]. \tag{4}$$

As per the DADA approach, we also added an entropy minimization loss to increase the mask decoder's confidence in its real domain predictions:

$$\mathcal{L}_{EM}(m_r) = \mathbb{E}_{n,c,h,w}[-m_r \log m_r]. \tag{5}$$

| Input | Depth | Segmentation | Mask | Masked Input | Painted Input |
|---|---|---|---|---|---|

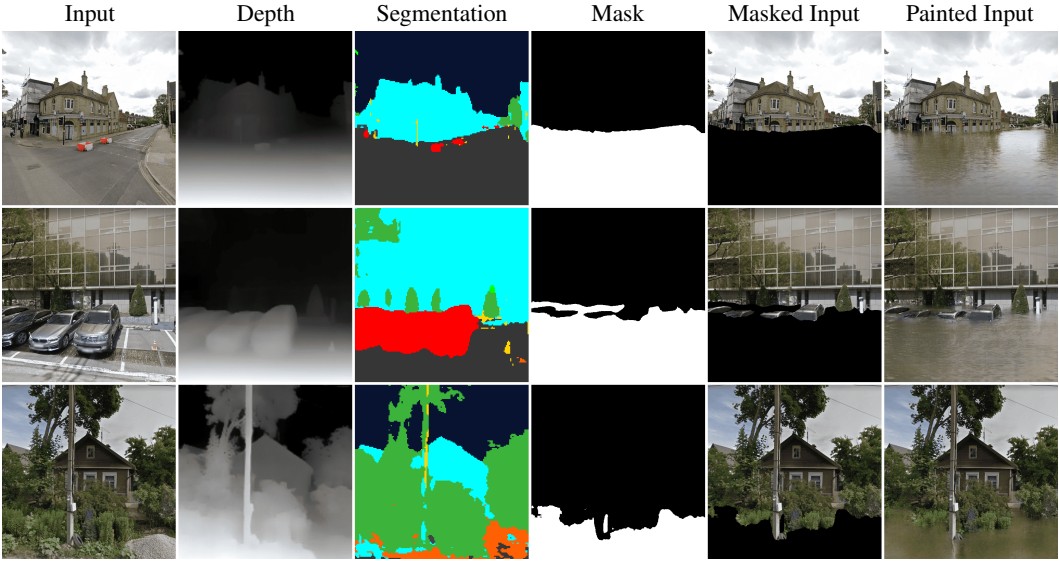

Figure 3: Example inferences of ClimateGAN, along with intermediate outputs. The first row shows how the Masker is able to capture complex perspectives and how the Painter is able to realistically contextualize the water with the appropriate sky and building reflections. On the second row, we can see that close-ups with distorted objects do not prevent the Masker from appropriately contouring objects. Finally, the last row illustrates how, in unusual scenes, the Masker may imperfectly capture the exact geometry of the scene but the final rendering by the Painter produces an acceptable image of a flood. More inferences including failure cases are shown in Appendix H.

Lastly, similarly to the segmentation decoder, we adversarially trained the flood mask decoder with a WGAN loss $\mathcal{L}_{WGAN}$ to produce self-information maps $\hat{I}(\boldsymbol{m}_i)$ indistinguishable by a discriminator. $M$'s total loss is a weighted sum of all of the above losses:

$$\mathcal{L}_{Mask} = \lambda_6 \mathcal{L}_{TV} + \lambda_7 \mathcal{L}_{GI} + \lambda_8 \mathcal{L}_{BCE} + \lambda_9 \mathcal{L}_{EM} + \lambda_{10} \mathcal{L}_{WGAN}. \tag{6}$$

The Masker's final loss sums the losses of the three decoders: $\mathcal{L}_{Masker} = \mathcal{L}_{Depth} + \mathcal{L}_{Seg} + \mathcal{L}_{Mask}$.

### 3.3 PAINTER

Given an input image and a binary mask, the goal of the Painter is to generate water in the masked area while accounting for context in the input image. This is important for the realism of a flooded image because water typically reflects the sky and surrounding objects, and is colored conditionally on the environment. The architecture of the Painter is based on GauGAN using SPADE conditioning blocks. These blocks learn transformations that vary both spatially and conditionally on the input, allowing the model to better propagate contextual information through the generation pipeline than if it were only available at the input of the generator. We adapted GauGAN to fit our task: rather than conditioning on a semantic map, we conditioned it on a masked image. Finally, we copied the non-masked area back onto the generated image, to ensure that the other objects in the image (e.g. buildings and sky) remained intact. We trained the Painter on the 1200 real flooded images, inferring $\boldsymbol{m}$ from pseudo labels of water segmented by a pre-trained DeepLabv3+ model. At test time $\boldsymbol{m}$ was generated by the Masker from $\boldsymbol{x}$. Thus, the output $\tilde{\boldsymbol{y}}$ of the Painter $P$ is:

$$\tilde{\boldsymbol{y}} = P(\epsilon, (1 - \boldsymbol{m}) \odot \boldsymbol{x}) \odot \boldsymbol{m} + \boldsymbol{x} \odot (1 - \boldsymbol{m}), \ \ \epsilon \sim N(\boldsymbol{0}, \mathbf{I}) . \tag{7}$$

According to Park et al. (2019), a perceptual VGG loss (Ledig et al., 2017) and a discriminator feature-matching loss (Salimans et al., 2016) are essential for good performance. Since these last two losses rely on comparing input and generated output, we trained the Painter using images of floods, separately from the Masker (which requires non-flooded images as inputs).

# 4 EVALUATION METHOD

The quality and photo-realism of our model's output depend on both the accuracy of the Masker in determining a realistic area for the flood and the ability of the Painter to infill the input mask with water with a realistic texture matching the surrounding scene. In order to best understand the contribution of each component, we evaluated the Masker and the final flooded images separately.

## 4.1 MASKER EVALUATION

The lack of ground-truth data is not only an obstacle for training, but also for obtaining reliable evaluation metrics to assess the performance of our model. Therefore, in order to obtain metrics for evaluating the quality of the Masker and for comparing the individual contribution of the proposed components to its architecture, we manually labeled a test set of 180 images retrieved from Google Street View. We collected images of diverse geographical provenances, levels of urban development, and compositions (crowds, vehicles, vegetation types, etc.). We manually annotated every pixel of each image with one of three classes: (1) *cannot-be-flooded*—pixels higher than 1.5m above the ground level; (2) *must-be-flooded*—anything with height less than 0.5 m, and (3) *may-be-flooded*—all remaining pixels. We provide further details in Appendix C.1.

### 4.1.1 METRICS

We propose the following metrics to evaluate the quality of masks (more details in Appendix C.2):

**Error rate**  The perceptual quality of the generated images is highly impacted by the location and area of the predicted masks' errors. We found that both large gaps in *must-be-flooded* areas (i.e. false negatives, $FN$), and large sections of predicted masks in *cannot-be-flooded* areas (i.e. false positives, $FP$) account for low perceptual quality. Thus, we propose the error rate to account for the amount of prediction errors in relation to the image size: $error = (FN + FP)/(H \times W)$.

**F05 Score**  So as to consider the precision and recall of the mask predictions, we also evaluate the $F_{\beta=0.5}$ (F05) score, which lends more weight to precision than to recall (van Rijsbergen, 1979).

**Edge coherence**  It is also important to take into account the *shape similarity* between the predicted mask and the ground-truth label, particularly in the uncertainty region defined by the *may-be-flooded* class. We capture this property by computing the standard deviation $\sigma$ of the minimum of Euclidean distances $d(\cdot, \cdot)$ between every pixel in the boundary of the predicted mask $\hat{B}$ and the boundary of the *must-be-flooded* area $B$, which are computed by filtering the masks with the Sobel edge detector:

$$edge\ coherence = 1 - \sigma \left( \min_j \left[ d(\hat{B}_i, B_j)/H \right] \right). \tag{8}$$

### 4.1.2 ABLATION STUDY

In order to assess the contribution of each of the components described in Section 3.2 to the overall Masker performance, we performed an ablation study by training 18 models, each with a different

| | 1 | 2 | 3 | 4 | 5 | 6 | 7 | 8 | 9 | 10 | 11 | 12 | 13 | 14 | 15 | 16 | 17 | 18 | G | I |
|---|---|---|---|---|---|---|---|---|---|---|---|---|---|---|---|---|---|---|---|---|
| Pseudo labels | • | • | • | • | • | • | • | • | • | | | | | | | | | | | |
| Depth ($D$) | | • | | • | • | • | • | • | • | | • | | • | • | • | • | • | • | | • |
| Segmentation ($S$) | | | • | | • | • | • | • | • | | | • | • | • | • | • | • | • | | • |
| SPADE | | | | | • | | | • | | | | | | | • | | • | | | |
| DADA ($S$) | | | | | | | • | • | | • | | | | | • | • | | • | | |
| DADA ($M$) | | | | | | | | • | • | | | | | | | | • | • | | |

Table 1: Summary of the ablation study of the Masker. Each numbered column corresponds to a trained model and black dots indicate which techniques (rows) were included in the model. The last two columns correspond to the baseline models: ground (G) segmentation from HRNet as flood mask and InstaGAN (I).

| Input | CycleGAN | MUNIT | InstaGAN | InstaGAN+Mask | Painter+Ground | ClimateGAN |

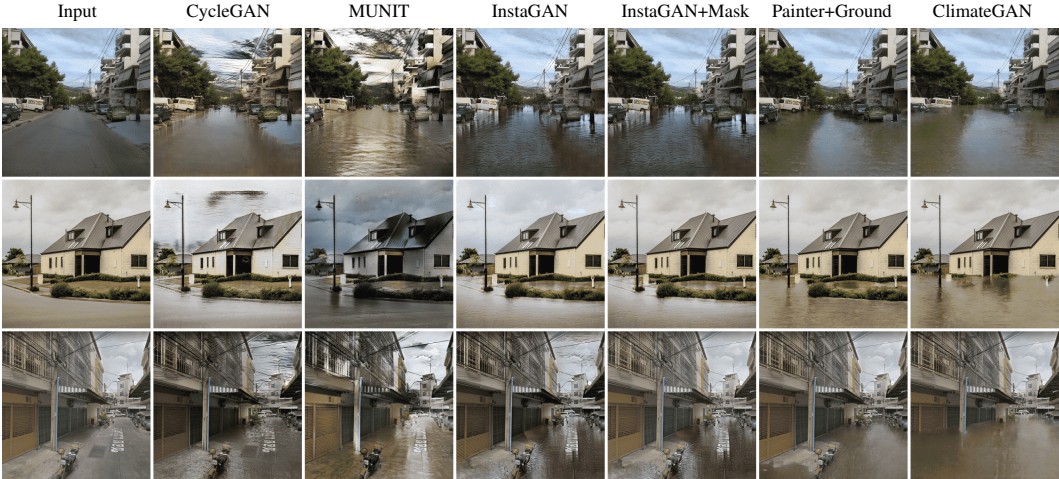

Figure 4: Example inferences of ClimateGAN and comparable approaches on three diverse street scenes from the test set. We can see that ClimateGAN is able to generate both realistic water texture and color, as well as a complex mask that surrounds objects such as cars and buildings. Comparable approaches are often *too destructive*, producing artifacts in the buildings and sky (e.g. $1^{st}$ row of MUNIT) or *not destructive enough*, resembling more rain on the ground than high floods (e.g. $3^{rd}$ row of CycleGAN and $2^{nd}$ row of InstaGAN).

combination of techniques. As illustrated in Table 1, we analyzed the effect of including the following components in the Masker architecture: training with pseudo labels, $D$, $S$, DADA for $S$, DADA for $M$, and SPADE for $M$. When not using SPADE, $M$ is based on residual blocks followed by convolutional and upsampling layers, taking $z$ as input, and we can also inform it with depth according to the DADA procedure just like we do for $S$.

We compared these variants of the Masker with two baselines: ground segmentation from HRNet pre-trained on Cityscapes instead of the flood mask (G) and InstaGAN (I). For each model variant in the ablation study (column in Table 1) we computed mask predictions for each of the 180 test set images and the values of the three metrics: error, F05 and edge coherence. In order to determine whether each technique (row) improves the performance, we computed the difference between the metrics on each image for every pair of models where the only difference is the inclusion or exclusion of this technique. Finally, we carried out statistical inference through the percentile bootstrap method (Efron, 1992) to obtain robust estimates of the performance differences and confidence intervals. In particular, we obtained one million bootstrap samples (via sampling with replacement) for each metric to get a distribution of the bootstrapped 20 % trimmed mean, which is a robust measure of location Wilcox (2011). We then compared the distribution against the null hypothesis, which indicates no difference in performance (see Fig. 5). We considered a technique to be beneficial for the Masker if its inclusion reduced the error rate. In case of inconclusive error rate results, we considered an increase in the F05 score and finally in the edge coherence.

## 4.2 COMPARABLES FOR HUMAN EVALUATION

While the nature of our task is specific to our project, we can nonetheless benchmark ClimateGAN against IIT models. In fact, in earlier iterations of our project, we leveraged the CycleGAN architecture in order to achieve initial results (Schmidt et al., 2019), before adopting a more structured approach. Therefore, to be as comprehensive in our comparisons as possible, we trained the following five models on the same data as the ClimateGAN Painter and used the same test set for the comparison: CycleGAN, MUNIT, InstaGAN, InstaGAN using the mask to constrain the transformation to only the masked area (similarly to Eq. (7)), and the ClimateGAN Painter applied to ground segmentation masks predicted by HRNet. These models were trained using the original papers' configurations, with hyper-parameter tuning to be fair in our comparison. Some samples from the models can be seen in Fig. 4, and further results are provided in Appendix D, Fig. 16.

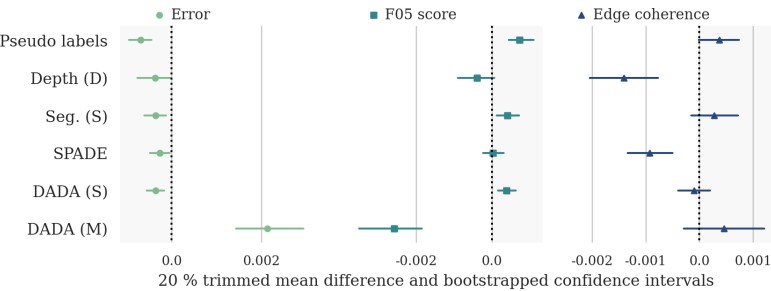

Figure 5: Statistical inference tests of the ablation study. Shaded areas indicate metric improvement. All techniques but DADA ($M$) significantly improved the error rate, and some further improved the F05 score and edge coherence.

## 5 RESULTS

This section presents the model evaluation results, including the Masker ablation study and the comparison of the overall proposed model—Masker and Painter—against comparable approaches via human evaluation. Visual examples of the inferences of our model can be seen in Fig. 3.

### 5.1 MASKER EVALUATION

The main conclusion of our ablation study is that five of the six techniques proposed to improve the quality of the Masker positively contribute to the performance. In Fig. 5, we show the median differences and confidence intervals obtained through the bootstrap. The error rate improved—with 99 % confidence—in the models that included pseudo labels, a depth head, a segmentation head, SPADE-based $M$ and DADA for the segmentation head. For some but not all techniques, the F05 score and edge coherence also improved significantly. In contrast, we found that both the error and the F05 score were worse when DADA for the Masker was included.

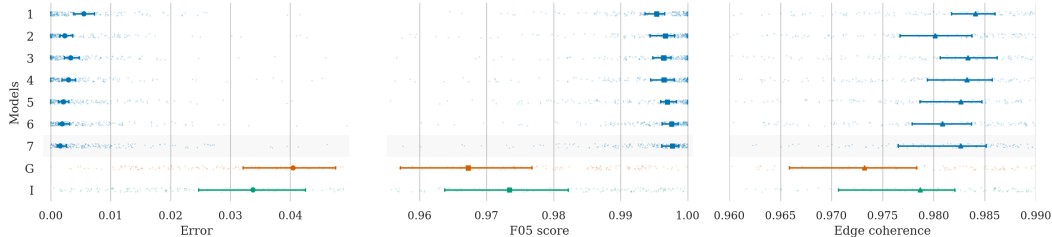

Figure 6: Evaluation metrics for a subset of the models studied in the ablation study and presented in Table 1—models trained without pseudo labels or with DADA for the Masker are excluded—as well as the two baselines for comparison. We show the median of the distribution with bootstrapped 99 % confidence intervals. The shaded area highlights the best model.

In Fig. 6, we show the Masker evaluation metrics for a subset of the models, 1–7, selected upon the conclusions of the ablation study, as well as the two baseline models. As main conclusion, our proposed Masker largely outperforms the baselines for the three metrics, especially in terms of the error and the F05 score. Further, the metrics of the individual models support the findings of the ablation study, as the best performance—5, 6 and 7—is achieved by the models that include all or most techniques: pseudo labels, depth and segmentation heads, DADA for $S$ and SPADE. Surprisingly, model 2, which only includes pseudo labels and the depth head, also achieves high performance. In contrast, models 3 and 4, which include a segmentation head without DADA or SPADE, obtain worse performance. Therefore, it seems that the contribution of the segmentation head is clearly complemented by using DADA and SPADE. A more complete analysis is provided in Appendix C. The final architecture we selected for the Masker includes: pseudo labels, $D$, $S$, DADA for $S$ and SPADE for $M$ (model 7 in Table 1).

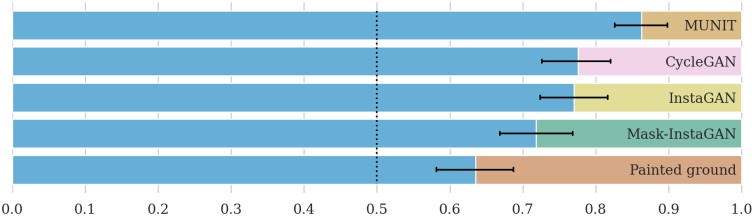

Figure 7: Results of the human evaluation: the blue bars indicate the rate of selection of Climate-GAN over each alternative. The error lines indicate 99 % confidence intervals.

## 5.2 Human Evaluation

So as to compare ClimateGAN to related alternatives, we asked human participants to select the image that looked more like an actual flood, given pairs in which one image was generated by ClimateGAN and the other by one of the models in Fig. 4. Participants chose ClimateGAN images most times in all cases (Fig. 7). Further details of this study are provided in Appendix D.

## 6 Future Work

An intuitive extension of our model would be to render floods at any chosen height. Despite the appeal of this approach, its development is compromised by data challenges. To our knowledge, there is no data set of metric height maps of street scenes, which would be necessary for converting *relative* depth and height maps into *absolute* ones. Moreover, simulated worlds—including our own—that have metric height maps do not cover a large enough range of scenes to train models that would generalize well on worldwide Google Street View images. Another promising direction for improvement would be to achieve multi-level flooding, controlling water level represented by a mask, which faces the same challenges.

We also explored the integration of several multi-task learning strategies to weigh the various Masker losses, including dynamic weight average (Liu et al., 2019) and weighting losses by uncertainty (Kendall et al., 2018). However, we empirically found that our manual tuning of constant weights performs better than the aforementioned methods. In future work, we aim to explore other techniques like gradient modulation methods (Yu et al., 2020; Maninis et al., 2019).

## 7 Conclusion

In this research, we have proposed to leverage advances in modern generative modeling techniques to create visualizations of floods, in order to raise awareness about climate change and its dire consequences. In fact, the algorithm described here has been incorporated into an interactive web-based tool that allows users to input any address of their choice and visualize the consequences of extreme climate events like floods, while learning about the causes and impacts of climate change[3].

Our contributions comprise a data set of images and labels from a 3D virtual world, a novel architecture ClimateGAN for the task of street-level flood generation, and a thorough evaluation method to measure the performance of our system. We found our proposed Masker-Painter dichotomy to be superior to existing comparable techniques of conditional generations in two ways: first, we empirically showed that we are able to produce more realistic flood masks by informing the flood mask decoder with geometrical information from depth predictions and semantic information from segmentation maps, and by training the three decoders together. Second, we established, using human evaluations, that our Painter alone is able to create realistic water textures when provided ground masks, and that its performance increases even further when provided with the Masker's predictions. Overall, this allows ClimateGAN to produce compelling and robust visualizations of floods on a diverse set of first-person views of urban, suburban and rural scenes.

---

[3]https://thisclimatedoesnotexist.com

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

## A    SIMULATED DATA

In this section, we expand on Section 3.1 and provide details about our simulated world and the images and labels we can obtain from it.

We created a 1.5 $km^2$ virtual world using the Unity3D game engine containing urban, suburban and rural areas.

The $urban$ environment contains skyscrapers, large buildings, and roads, as well as objects such as traffic items and vehicles. Fig. 8 shows a bird's eye view of the urban area of our virtual environment. The $rural$ environment consists of a landscape of grassy hills , forests, and mountains, with sparse houses and other buildings such as a church, and no roads. The rural and urban areas make up for 1 $km^2$ of our virtual world.

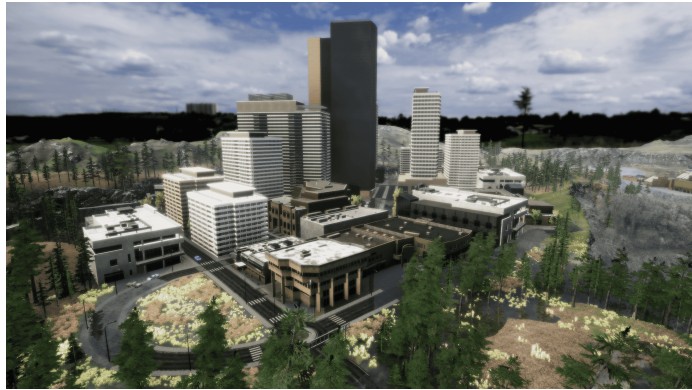

Figure 8: Bird's eye view of the urban area (city) and rural area (outskirts of the city) of our simulated world

The $suburban$ environment (Figure 9) is a residential area of 0.5 $km^2$ with many individual houses with front yards.

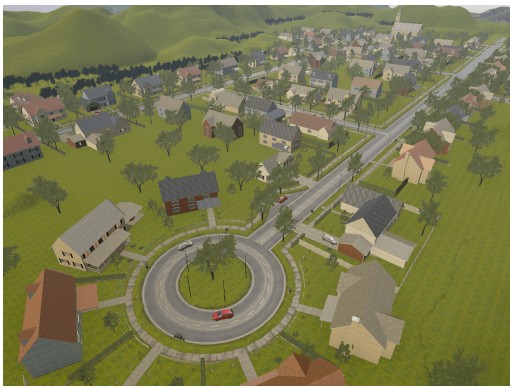 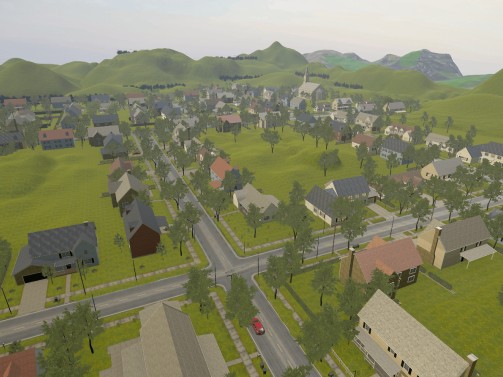

Figure 9: Bird's eye views of the suburban area of our simulated world

To gather the simulated dataset, we captured 'before' and 'after' (without/with) flood pairs from 2000 viewpoints with the following modalities:

- 'before' : non-flooded RGB image, depth map, segmentation map
- 'after' : flooded RGB image, binary mask of the flooded area, segmentation map

The camera was placed about $1.5m$ above ground, and has a field of view of $120°$, and the resolution of the images is $1200 \times 900$. At each viewpoint, we took 10 pictures, by varying slightly the position of the camera in order to augment the dataset.

Fig. 10 shows the different modalities captured at each viewpoint, and Fig. 11 shows samples of our simulated dataset in urban, suburban and rural areas.

**Depth**   The depth maps are provided as RGB images for the 'before' (without flood) case, and the depth is recorded up to $1000m$ away from the camera, with precision of $4mm$.

**Segmentation**   There are nine different classes of objects in the simulated world:

- *sky*
- *ground*: road, sidewalks, road markings, anything that is asphalt
- *building*
- *traffic item*: lampposts, traffic signs, poles
- *vegetation*: small bushes, trees, hedges excludes grass, lawns
- *terrain*: rocks, soil, lawns
- *car*: cars and trucks
- *other*: miscellaneous objects such as postboxes, trashcans, garbage bags, etc.
- *water*: only present in the 'after' flooded images

While people are not included in the simulated world, the segmentation model is able to learn this class from the real world due to the supervision signal given by the HRNet pseudo-labels.

**Mask**   We also include binary masks of the flood (water segmentation) for the 'after' images. The masks are used to train the Masker with ground truth target flood masks in the simulated domain.

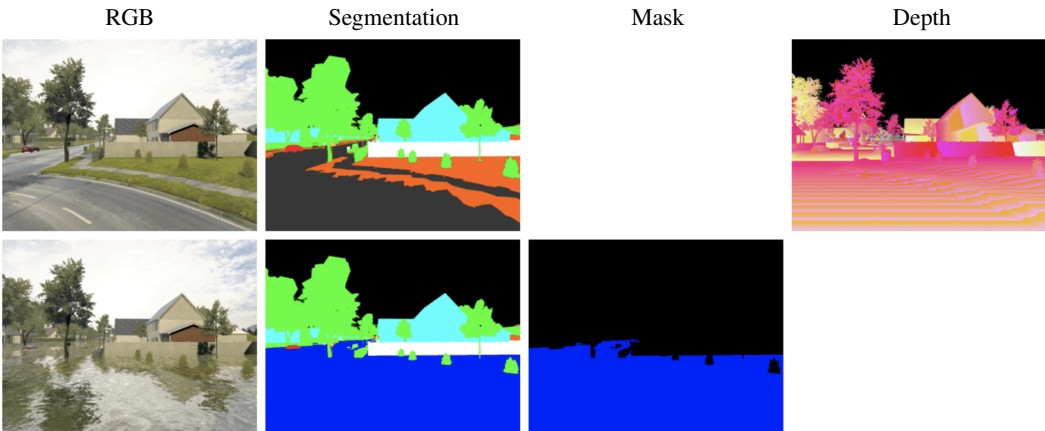

Figure 10: Sample data obtained at one spot for one camera position in our virtual world. The top row shows the modalities of the 'before' flood image: RGB image of the scene, depth map and segmentation map; and the bottom row shows those obtained in the 'after' configuration: RGB image, segmentation map and binary flood mask.

## B   LOSSES

For clarity, in Section 3.2 we did not detail the exact formulation of some of the losses since they are either straight-forward (as is the Cross Entropy) or direct applications of their definitions as per their original paper (*e.g.* the SSIMSE loss). We expand on those here. In particular, to guide decoders, we used *pseudo labels* from pre-trained models as supervision targets for real data. Because these labels are noisy, we limited this procedure to the first ten epochs of training.

### B.1   DEPTH DECODER

In this section, we detail the depth decoder loss presented in Section 3.2 and adapted from (Lasinger et al., 2019).

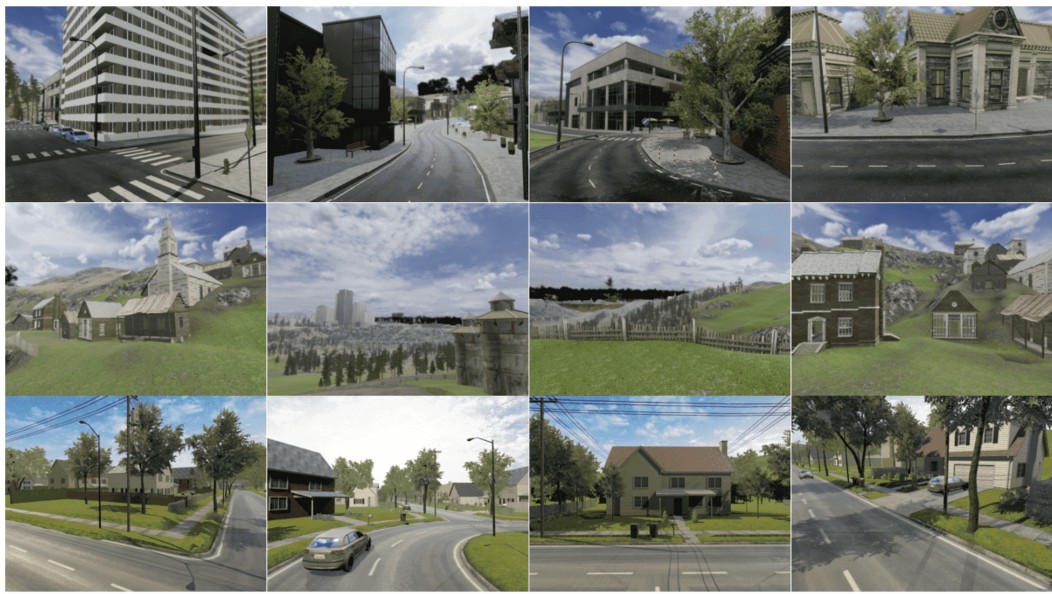

Figure 11: Samples from our simulated dataset in urban (top row), rural (middle row), and suburban (bottom row) areas.

Let $d$ a predicted disparity map and $d^*$ the corresponding ground truth. The aligned disparity maps with zero translation and unit scale are:

$$\hat{d} = \frac{d - \mathbf{t}(d)}{\mathbf{s}(d)} \;,\; \hat{d}^* = \frac{d^* - \mathbf{t}(d^*)}{\mathbf{s}(d^*)} \tag{9}$$

where $\mathbf{t}$ and $\mathbf{s}$ are defined as $\mathbf{t}(d) = \text{median}(d)$ and $\mathbf{s}(d) = \frac{1}{N}\sum_n |d - \mathbf{t}(d)|$.

The depth decoder loss is composed of a scale-and-shift invariant MSE loss term :

$$\mathcal{L}_{SSIMSE} = \frac{1}{2}\, \mathbb{E}_{n,h,w}[(\hat{d}^{(n,h,w)} - \hat{d}^{*(n,h,w)})^2] \tag{10}$$

an a multi-scale gradient matching term to enforce smooth gradients and sharp discontinuities:

$$\mathcal{L}_{GM} = \mathbb{E}_n[\sum_k \sum_{h,w} |\nabla_x R_k^{(n,h,w)}| + |\nabla_y R_k^{(n,h,w)}|] \tag{11}$$

where $R^{(n,h,w)} = \hat{d}^{(n,h,w)} - \hat{d}^{*(n,h,w)}$ and $R_k$ corresponds to the difference of disparity (inverse depth) maps at scale $k$.
Following (Lasinger et al., 2019), we consider 4 scale levels, downsampling by a factor two the image at each scale.

### B.2 SEGMENTATION DECODER

This decoder computes the segmentation map $s_i$ given an input image $x_i$, $s_i$ being a 4D tensor of shape $N \times C \times H \times W$. The number of channels $C$ corresponds to the number of classes, nine in our case: *ground*, *building*, *traffic item*, *vegetation*, *terrain*, *car*, *sky*, *person* and *other*.

First, we detail the two fusion mechanisms used in the DADA approach (Vu et al., 2019b):

- **Feature fusion**: It is the element-wise multiplication between the latent vector and a depth vector of the same size. This depth vector is obtained by a 1 x 1 convolutional layer applied to the depth head before the average pooling. The fused features are given as input to the segmentation decoder.

- **DADA fusion**: Instead of giving the self-information map $I_i^S$ to the AdvEnt (Vu et al., 2019a) discriminator $Q^S$, we give it the element-wise multiplication between $I_i^S$ and the depth predictions $\boldsymbol{d}_i$. The obtained matrix is called the depth-aware map $\hat{I}_i^S$.

We also detail all segmentation losses that are not defined in the main paper. First, the cross-entropy loss is defined as:

$$\mathcal{L}_{CE}(\boldsymbol{y}_i, \boldsymbol{s}_i) = -\sum_c \mathbb{E}_{n,h,w}[\boldsymbol{y}_i^{(n,c,h,w)} \log \boldsymbol{s}_i^{(n,c,h,w)}] \tag{12}$$

Furthermore, the entropy minimization loss, which is used as suggested by the authors of AD-VENT(Vu et al., 2019a), is computed according to the following equation:

$$\mathcal{L}_{EM}(\boldsymbol{s}_r) = \mathbb{E}_{n,c,h,w}[-\boldsymbol{s}_r \log \boldsymbol{s}_r] \tag{13}$$

Finally, we detail the WGAN loss that is computed using a discriminator $Q_s$ which outputs the probability of the depth-aware map coming from the real domain. If updating the decoder $S$, the loss tries to fool the discriminator:

$$\mathcal{L}_{G-GAN}(Q^S, \hat{I}(\boldsymbol{s}_r), \hat{I}(\boldsymbol{s}_s)) = -\mathbb{E}_n[Q^S(\hat{I}(\boldsymbol{s}_r))] \tag{14}$$

If updating D, the loss tries to improve the discriminator's prediction:

$$\mathcal{L}_{D-GAN}(Q^S, \hat{I}(\boldsymbol{s}_r), \hat{I}(\boldsymbol{s}_s)) = -\mathbb{E}_n[Q^S(\hat{I}(\boldsymbol{s}_s)) - Q^S(\hat{I}(\boldsymbol{s}_r))] \tag{15}$$

### B.3 FLOOD MASK DECODER

Similarly to the segmentation decoder, we detail in this section the WGAN loss that is computed using a discriminator $Q^M$ in order to train M. This discriminator outputs the probability of the self-information map $I(\boldsymbol{m}_i)$ coming from the real domain. When updating the decoder $M$, the loss tries to fool the discriminator:

$$\mathcal{L}_{G-GAN}(Q^M, I(\boldsymbol{m}_r), I(\boldsymbol{m}_s)) = -\mathbb{E}_n[Q^M(I(\boldsymbol{m}_r))] \tag{16}$$

When updating $Q^M$, the loss tries to improve the discriminator's predictions:

$$\begin{aligned}\mathcal{L}_{D-GAN}(Q^M, I(\boldsymbol{m}_r), I(\boldsymbol{m}_s)) = \\ -\mathbb{E}_n[Q^M(I(\boldsymbol{m}_s)) - Q^M(I(\boldsymbol{m}_r))]\end{aligned} \tag{17}$$

## C MASKER EVALUATION

In Section 4.1, we introduced the method for evaluating the quality of Masker's output. Here, we provide details of the annotation procedure for generating a Masker test set (Appendix C.1) and the metrics proposed to assess the performance (Appendix C.2).

### C.1 TEST SET COLLECTION AND ANNOTATION

In this section, we provide more details about our test set data collection procedure, and the annotation guidelines we followed.
We collected the 180 images of our test set from Google Street View with the following selection guidelines:

- Geographical diversity: we selected images in a variety of cities on all continents, with different architectural styles and general landscapes.

- Varied levels of urban development: both urban and rural areas were included.
- Variety of scenes content: we endeavored to cover a wide range of objects in our test set, including images containing bike racks, dense crowds, and various vehicles in urban areas, and different vegetation types in rural areas.

We also carefully collected challenging images to determine the limitations of our model, including images facing slopes (going up or down), or stairs, with various ground textures, around areas under construction, and in areas near canals among others. The images were manually annotated, and the pixels of each image were categorized in one of three classes according to the following instructions:

- *Must be flooded*: This class contains the minimal region that should be flooded. We want to represent floods of height at least 0.5 m. Typically, this corresponds to flooding up to the knees of adult pedestrians or up to the top of cars' wheels. In cases when no such reference objects were available, the annotator would make the 0.5m estimate based on other cues, such as doorsteps, traffic signs and vegetation.
- *Cannot be flooded*: This label indicates regions we absolutely do not want to be flooded. We generally put any pixel corresponding to an object with a height greater than 1.5 m above ground in this category. This includes car roofs and adult pedestrians' heads, which we used as reference to determine the lower limit of this region.
- *May be flooded*: This category contains any pixel not assigned to the other classes. It reflects the fact that we do not enforce flooding at a specific height yet value plausible flood location.

We show examples of the labeled test images in Fig. 12.

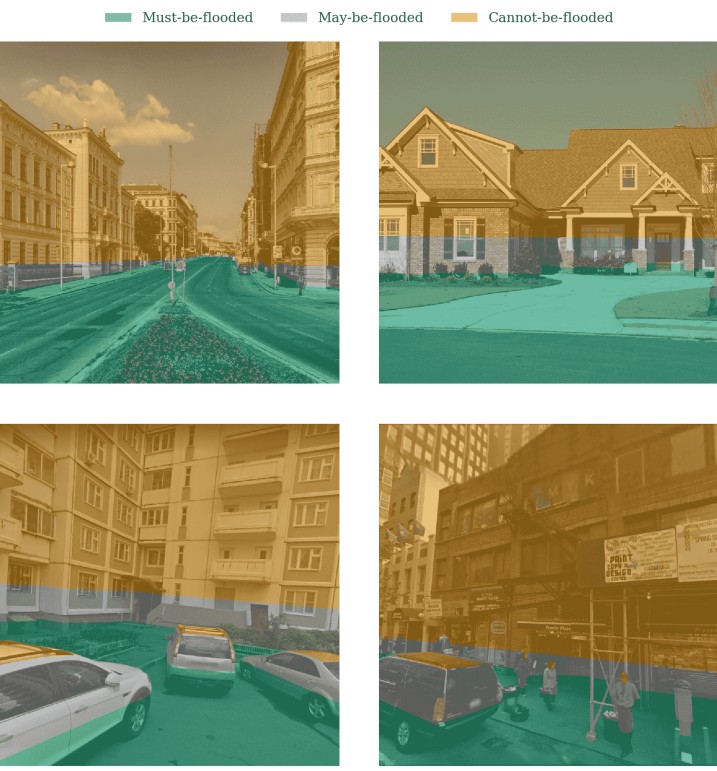

Figure 12: Examples of labeled images from our test set.

C.2   METRICS

In Section 4.1.1 we proposed three metrics to compare the masks predicted by the Masker and the labeled test images: error, F05 score and edge coherence (Eq. (8)). Here, we delve into the reasons why we proposed more than one metric and the contribution of each metric.

We proposed the error—number of erroneously predicted pixels divided by the size of the image—as the main evaluation metric, since it characterizes the size of the errors in the image, which directly impacts the perceived mask quality.

However, we argue that the error alone does not capture all aspects of the performance of a Masker model in the test set. For example, the error does not take into account the size of the labeled areas (see image "D" in Fig. 19). While the size of the labels may have a smaller perceptual impact, in order to characterize the precision and sensitivity of the model, additional metrics are useful. We proposed the F05 score:

$$F05 = \frac{1.25 \times precision \times recall}{0.25 \times precision + recall} = \frac{1.25 \times \frac{TP}{TP+FP} \times \frac{TP}{TP+FN}}{0.25 \times \frac{TP}{TP+FP} + \frac{TP}{TP+FN}} \tag{18}$$

which computes the weighted harmonic mean of precision and recall—also known as sensitivity or true positive rate. We used $\beta = 0.5$ in order to weigh precision more than recall, that is to penalize more false positives than false negatives. In our context, this translates into setting a higher penalty for flooding *cannot-be-flooded* pixels—for instance heads of pedestrians, automobile roofs and high areas of the image, in general—than for missing areas that should be flooded. While both types of errors should be penalized, the former has a higher perceptual impact.

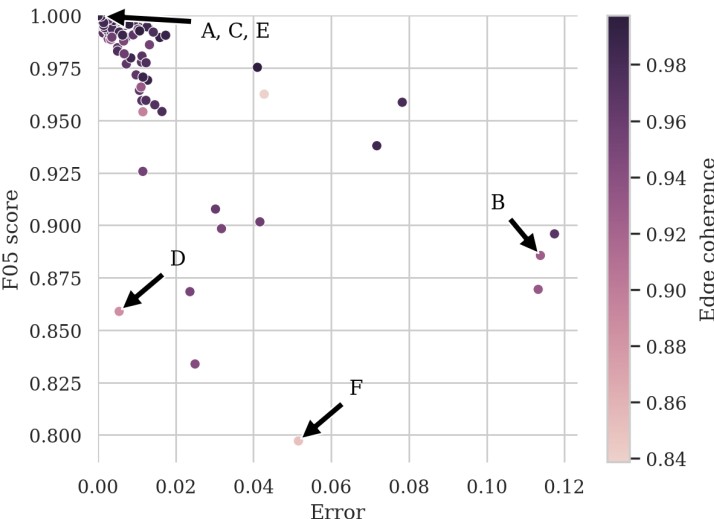

Figure 13: Distribution of the three metrics for the selected best Masker. The annotations correspond to the images in Fig. 19.

Finally, we proposed an edge coherence metric (Eq. (8)) in order to take into consideration the shape of the predicted mask, with respect to the shape of the *must-be-flooded* label. Note that neither the error or the F05 score account for the shape, as only the amount of correct or incorrect pixels matter, regardless of the position. As explained in the previous section, we defined a *may-be-flooded* class in order to allow for different levels of flooding in the mask prediction. That is, higher levels of flooding in a prediction should not be penalized necessarily, as long as the mask is consistent with the semantics in the image. Our proposed metric of edge coherence is based on the assumption that predictions whose border is roughly parallel to the border of the *must-be-flooded* label should be less penalized than highly dissimilar shapes.

Fig. 13 shows the value of the three metrics for all the images in the test set. While there is certain correlation between the metrics, especially because the bulk of the distributions is around images for

| | Error | p |
|---|---|---|
| Pseudo labels | $-6.9 \times 10^{-4}$ $[-9.5 \times 10^{-4}, -4.5 \times 10^{-4}]$ | $= 0.0$ |
| Depth | $-3.7 \times 10^{-4}$ $[-7.3 \times 10^{-4}, -1.4 \times 10^{-5}]$ | $<0.01$ |
| Seg. (S) | $-3.6 \times 10^{-4}$ $[-6.0 \times 10^{-4}, -1.2 \times 10^{-4}]$ | $<0.0001$ |
| SPADE | $-2.6 \times 10^{-4}$ $[-4.7 \times 10^{-4}, -5.1 \times 10^{-5}]$ | $<0.01$ |
| DADA (S) | $-3.5 \times 10^{-4}$ $[-5.5 \times 10^{-4}, -1.7 \times 10^{-4}]$ | $= 0.0$ |
| DADA (M) | $+2.1 \times 10^{-3}$ $[+1.5 \times 10^{-3}, +2.9 \times 10^{-3}]$ | $= 0.0$ |

Table 2: Details of the results of the ablation study. The values in each cell are the 20 % trimmed mean *error* difference between models with and without a given technique, in brackets the 99 % confidence intervals, and the $p$ value.

| | F05 score | p |
|---|---|---|
| Pseudo labels | $+7.4 \times 10^{-4}$ $[+4.3 \times 10^{-4}, +1.1 \times 10^{-3}]$ | $= 0.0$ |
| Depth | $-3.9 \times 10^{-4}$ $[-9.1 \times 10^{-4}, +5.9 \times 10^{-5}]$ | $<0.1$ |
| Seg. (S) | $+4.1 \times 10^{-4}$ $[+1.2 \times 10^{-4}, +7.3 \times 10^{-4}]$ | $<0.001$ |
| SPADE | $+2.9 \times 10^{-5}$ $[-2.6 \times 10^{-4}, +2.9 \times 10^{-4}]$ | $>0.01$ |
| DADA (S) | $+3.8 \times 10^{-4}$ $[+1.3 \times 10^{-4}, +6.5 \times 10^{-4}]$ | $<0.0001$ |
| DADA (M) | $-2.6 \times 10^{-3}$ $[-3.5 \times 10^{-3}, -1.8 \times 10^{-3}]$ | $= 0.0$ |

Table 3: The analogue to Table 2 for the F05 score

which the predictions are very accurate and hence all the metrics are near perfect, the plot shows that various metrics are useful for identifying a few images for which not all metrics are low. We further illustrate the meaning of our proposed metrics in Fig. 19, where we show images that obtained the lowest—2nd quantile—and highest—98th quantile—values of each metric.

## C.3 ABLATION STUDY

Here, we detail the methodology used for the ablation study of the Masker, presented in Section 4.1.2, as well extend the set of results provided in Section 5.1.

In the ablation study, we studied the contribution to the Masker evaluation metrics of each technique: training with pseudo labels, a depth head ($D$), a segmentation head ($S$), SPADE, DADA for the segmentation head and DADA for the masker head. For every technique $t$, we considered all models $m_j^t$ which included such technique, and the paired models $m_j^{t_0}$ which differed only by the absence of $t$. Then, for every metric $r$, we constructed datasets of metric differences for every image $i$:

$$d_{ij}^r = r(m_j^t)_i - r(m_j^{t_0})_i \tag{19}$$

and obtained 1 million bootstrapped samples. On each bootstrap sample we computed the 20 % trimmed mean, which forms a bootstrap distribution from which we derived the confidence intervals

| | Edge coherence | p |
|---|---|---|
| Pseudo labels | $+3.7 \times 10^{-4}$ $[+1.2 \times 10^{-5}, +7.4 \times 10^{-4}]$ | $<0.01$ |
| Depth | $-1.4 \times 10^{-3}$ $[-2.1 \times 10^{-3}, -7.4 \times 10^{-4}]$ | $= 0.0$ |
| Seg. (S) | $+2.8 \times 10^{-4}$ $[-1.7 \times 10^{-4}, +7.3 \times 10^{-4}]$ | $>0.01$ |
| SPADE | $-9.3 \times 10^{-4}$ $[-1.4 \times 10^{-3}, -5.0 \times 10^{-4}]$ | $= 0.0$ |
| DADA (S) | $-1.0 \times 10^{-4}$ $[-4.2 \times 10^{-4}, +2.2 \times 10^{-4}]$ | $>0.01$ |
| DADA (M) | $+4.6 \times 10^{-4}$ $[-2.8 \times 10^{-4}, +1.2 \times 10^{-3}]$ | $>0.01$ |

Table 4: The analogue to Table 2 for the edge coherence

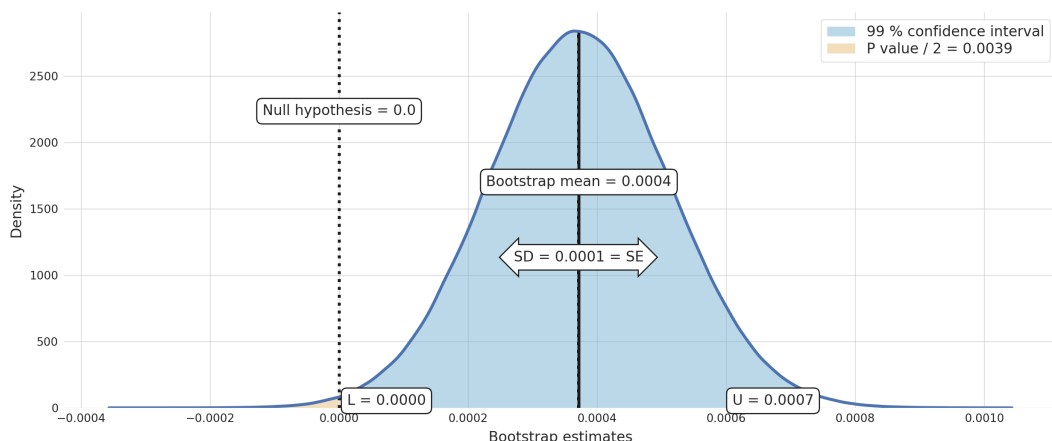

Figure 14: Bootstrapped distribution of the 20 % trimmed means of the difference in edge coherence between models that included pseudo labels and their counterparts. Equivalent distributions were obtained for all other techniques and metrics in the ablation study.

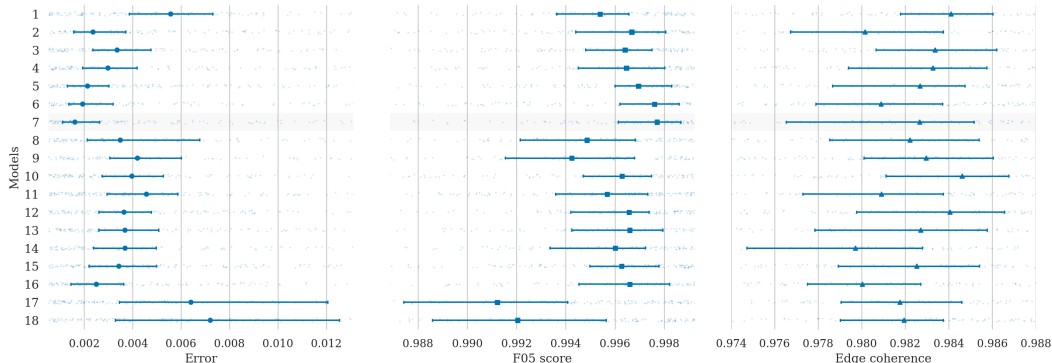

Figure 15: Distribution of the metrics—error, F05 score and edge coherence—of the complete set of models tested in the ablation study, 1–18, excluding the two baselines, whose distributions are shown in Fig. 6. The solid symbols indicate the median of the distribution and the error lines the bootstrapped 99 % confidence intervals. The shaded area highlights the best model—7.

we reported graphically in Fig. 5. In particular, we computed the 99 % confidence intervals, that is the lower bound is the 0.5th quantile and the upper bound is the 99.5th quantile of the bootstrap distribution. In Table 2, we provide the numerical details of all the tests in the ablation study. One advantage of the bootstrap over other statistical inference methods is that the outcome is a data-driven distribution rather than a binary test. In order to illustrate this, we provide the distribution for one of the techniques—pseudo labels—and one of the metrics—edge coherence—in Fig. 14.

In Fig. 15 we extend Fig. 6, where we show the distribution of the metrics for the complete set of models in the ablation study. We do not include the baseline models, which obtained significantly worse metrics, in order to better visualize the differences within the models of the ablation study. From the figure, it becomes apparent that models trained without pseudo labels (9–18), and models trained with DADA for the masker head (8, 9, 17, 18) achieved worse performance, in general. This is consistent with the results of the ablation study obtained through the bootstrap.

## D  HUMAN EVALUATION RESULTS

147 evaluators took part in the study (done via the Mechanical Turk platform) and we gathered 2700 total evaluations, 3 for each pair of images. For each image generated by ClimateGAN, we com-

pared it to 5 other approaches (MUNIT, InstaGAN, CycleGAN, InstaGAN with Mask and Painted Ground), and each one of these pairs was seen by 3 evaluators.

During the human evaluation of our results, we found that overall, evaluators primarily preferred ClimateGAN to other comparable approaches to varying degrees (a complete presentation of results can be found in Section 5.2). However, there were a number of cases when evaluators preferred other approaches to ClimateGAN. We present such images in Fig. 16, where the images outlined in red are those that are those for which all 3 evaluators preferred a comparable approach to ClimateGAN.

It can be observed that it is hard to consider these approaches as being *better* than ClimateGAN – i.e., they do not appear to be more realistic neither in terms of the quality of generated water nor in the portion of the image that is flooded. However, the concept of realism and quality of generated imagery is, indeed, hard to define and to quantify, so it is unsurprising that some images were systematically preferred by evaluators. In order to carry out a more thorough human evaluation of our results, it would be necessary to have more evaluators per pair of images and, ideally, different experimental setups – for instance, different captions and different designs of the comparison interface. Overall, it is hard to draw a conclusion for the motivation behind the preferences of individual evaluators. It can nonetheless be observed that in the case of some models, such as InstaGAN, images of light flooding (i.e. that after moderate rain) were preferred to that of more severe flooding generated by ClimateGAN. In other cases, such as that of InstaGAN+Mask, images with more pronounced reflections were chosen over those of the murkier, rippled water produced by ClimateGAN.

Finally, we feel that the best evaluation of the *impact* of images generated by ClimateGAN is to compare them with other mediums of climate communication. For this purpose, we are working with a group of researchers in psychology to test the effect that our images have on climate risk perception. The experimental setup involves providing subjects with a text regarding the risks of climate change-induced flooding. One group of subjects will only see the text, a second will see the text accompanied by an image of a generic flooded house, whereas the third will see an image of their address of residence flooded using ClimateGAN. Preliminary results have indicated that individuals who saw their own place of residence flooded are more likely to perceive climate change as a real threat to their livelihood, and more likely to take action to fight it.

## E  MORE DISCUSSION

### E.1  FAILURE CASES

The Painter seems to systematically generate realistic water textures, successfully incorporating the context of the scene such as reflections of buildings on the water, provided the mask is reasonably accurate. In other words, we have not identified failure patterns of the Painter, specifically. What's more, the output of the Painter often mitigates small failures in the accuracy of the mask: even when the mask is not perfect, the image output by the Painter may look perceptually realistic. The main source of failures, where the final image does not look perceptually realistic, is instead the Masker. This is expected given the more challenging nature of the task—predicting a mask for a hypothetical flood, taking into account the geometry of the scene, semantics, etc. The following are the kinds of images where we have found the Masker to perform sub-optimally at times:

- Images where the sky or the ground are not visible
- Images with a large object, such as a car or a truck, taking a large fraction of the foreground
- Wide open views of grass fields - the edges of the water mask are sometimes incorrect
- Scenes with multiple people in the scene - the edges of the water mask around all objects are sometimes inaccurate, although the perceptual error is mitigated by the Painter
- Scenes with much vegetation, such as bushes
- Urban scenes with a pronounced slope

Surprisingly, the painted image sometimes looks realistic in these cases, despite an inaccurate Mask. We hypothesize that the reason for lower accuracy of the Masker in these cases is the lack of such images in the training set, beside the additional difficulty intrinsic to some of these images, even to a human annotator.

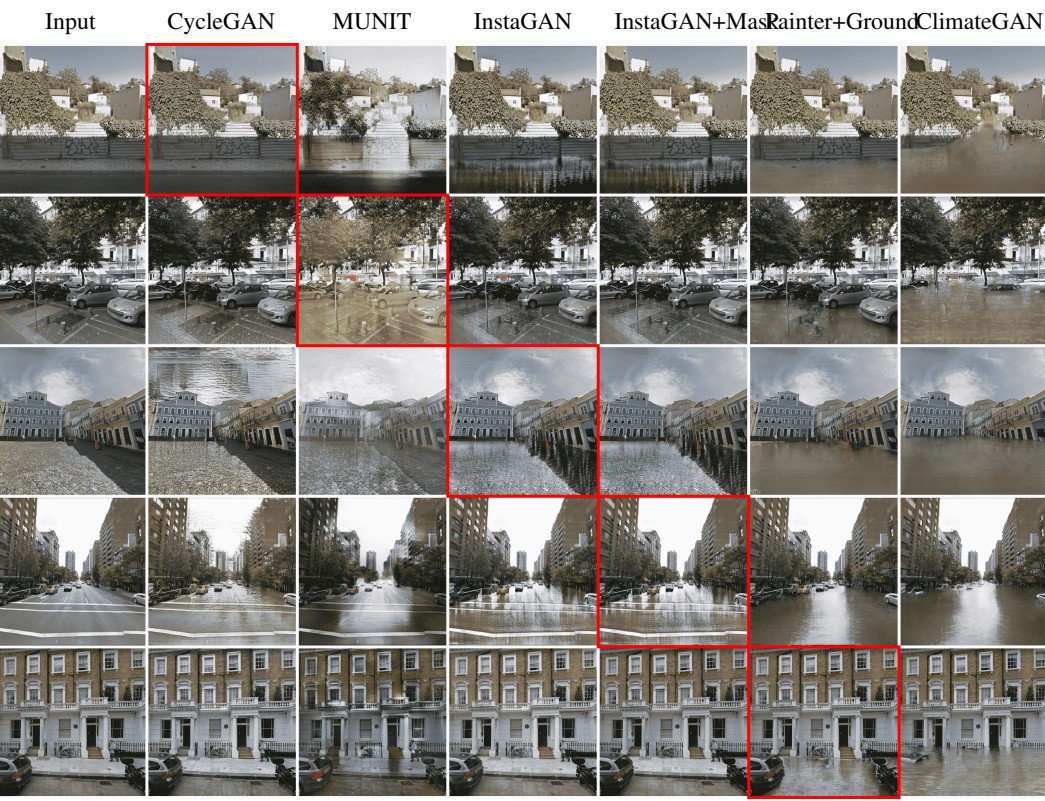

Figure 16: Example inferences of ClimateGAN and comparable approaches on images from the test set. Images outlined in red are those that were systematically (i.e. 3 out of 3 times) preferred over ClimateGAN.

### E.2 END-TO-END ARCHITECTURE

We studied the possibility to train the Masker and the Painter jointly. One major constraint to it is that two of the Painter's losses (Feature Matching Loss and Perceptual Loss) require the input data and generated data to be compared, and should therefore come from the same "category": flooded scenes. In other words, both those losses compare the input and the output to encourage the Painter to produce more realistic water. On the other hand, the Masker never processes such data: it produces masks from non-flooded images. Thus, we tried the 2 following approaches:

- Remove the two aforementioned losses from the Painter's training procedure, and train it jointly with the masker from non-flooded data

$$\tilde{y}(x) = P(\epsilon, (1 - \text{Masker}(x)) \odot x) \odot \text{Masker}(x) + x \odot (1 - \text{Masker}(x)) \qquad (20)$$

  As we expected, this procedure not only makes training the masker more difficult but we could not make the Painter to produce realistic water.

- Keep the two losses, train the Painter and Masker from different data sources (flooded/non-flooded) but use the loss from the Discriminator of the Painter to train the Masker. In other words, train $P$ and $M$ as described in the main body but add a loss ($P$ is frozen here) $L_{GAN_P}(D_P(\tilde{y}(x)))$ where $L_{GAN_P}$ is a GAN loss (cross-entropy or $L_2$ for instance), and $\tilde{y}$ is defined above. Since the Discriminator $D_p$ in the Painter's adversarial training procedure is supposed to assess how "realistic" its output is, then this takes into account not only how the water is painted, but also where it is. Unfortunately this proved to be very challenging in terms of computational complexity, memory footprint and most of all, convergence. We did not find a proper scheduling/scaling mechanism for $L_{GAN_P}$ to be informative enough to improve the Masker's performance.

### E.3 CLIMATEGAN VS ALL EVALUATION

In the human evaluation section, readers may be interested in 1-vs-all comparisons, comparing ClimateGAN outputs to all comparable methods at once. However, such a procedure would introduce methodological complications, both in the statistical tools at hand and because of the biases known to affect human evaluators when performing multi-way comparisons. For instance, laterality is well known for largely impacting visual inspection (Ossandón et al., 2014) and two-alternative forced choices is known to be an easier and more reliable task for human evaluators, which is why it is the common approach to measure sensitivity to stimuli or preferences in the cognitive sciences (Link & Heath, 1975). In addition, recent research in GAN evaluations also recommends making pairwise comparisons in order to measure user preference (Zhou et al., 2019). Furthermore, in our case we are mainly interested in knowing whether the images produced by our model are preferred over other alternatives, rather than comparing these alternatives against each other.

## F  CARBON IMPACT

The environmental impact of machine learning is becoming an increasingly major issue for our field, given the extensive experimentation and hyper-parameter tuning required to successfully train large neural networks (Strubell et al., 2019; Schwartz et al., 2019; Lacoste et al., 2019). In order to estimate our own carbon footprint, we counted all of the GPU hours (weighted by GPU usage) that our entire team used during the whole course of our project, from initial stages to model selection to the ablation study and hyper-parameter tuning on our final ClimateGAN model. We used the MLCO$_2$ Emissions Calculator[4] (Lacoste et al., 2019) to obtain a final estimate of **362.72 kilograms of CO$_2$eq.**, which is comparable to 900 miles driven by an average passenger vehicle, 42% of a US household's yearly energy consumption [5] or round trips between Paris, France and Saint-Petersburg, Russia (369kg CO$_2$eq.) or between Boston, MA and Miami, FL (345kg CO$_2$eq) [6]. This does not include the rest of the computational infrastructure's energy consumption (CPUs, data transfers and storage *etc.*) nor the full Life Cycle Analysis of any of the hardware used.

---

[4]mlco2.github.io/impact
[5]Source: EPA
[6]Source: ICAO

We are incredibly lucky that our power grid is powered predominantly by renewable energy. If this were not the case and we were using coal-powered energy, the figure stated above would be 50-80 times larger, and thereby more problematic. We hope that our colleagues will also start tallying and sharing the carbon footprint of their research using tools such as CodeCarbon and the Experiment Impact Tracker and that our community will start being more mindful regarding the trade-off between scientific progress and environmental impact.

## G    FULL CLIMATEGAN ARCHITECTURE

In order to better illustrate the overall training procedure of ClimateGAN, we provide a more detailed overview in Fig. 17.

The Masker can be seen in the top part of the Figure, encompassing the three decoders described in Section 3.2: Depth (D) , Segmentation (S) and Flood-Mask (M). The Painter is shown in the lower part of the image, with its SPADE-based encoder P. The losses of each component are indicated in the rounded white boxes within each decoder. Tensors are represented using squares, with different colors for input, label, intermediate and output tensors, and pseudo-labels with dotted outlines. To emphasize the SPADE-based conditional architectures of M and P, they are colored in green. As per Fig. 2, the output of ClimateGAN, *i.e.* the flooded image $\tilde{y}$, is in blue. Note that the dotted and dashed line from $m_r$ to P is not leveraged during training and only used for test-time inferences. Discriminators are conceptually included in the WGAN losses.

## H    SUPPLEMENTARY IMAGES

In order to further illustrate the performance and capabilities of ClimateGAN, we provide additional inferences in Fig. 18, complementing those presented in Fig. 3. The first two rows were chosen to be successful inferences, middle two rows were selected at random and bottom two rows are failure cases.

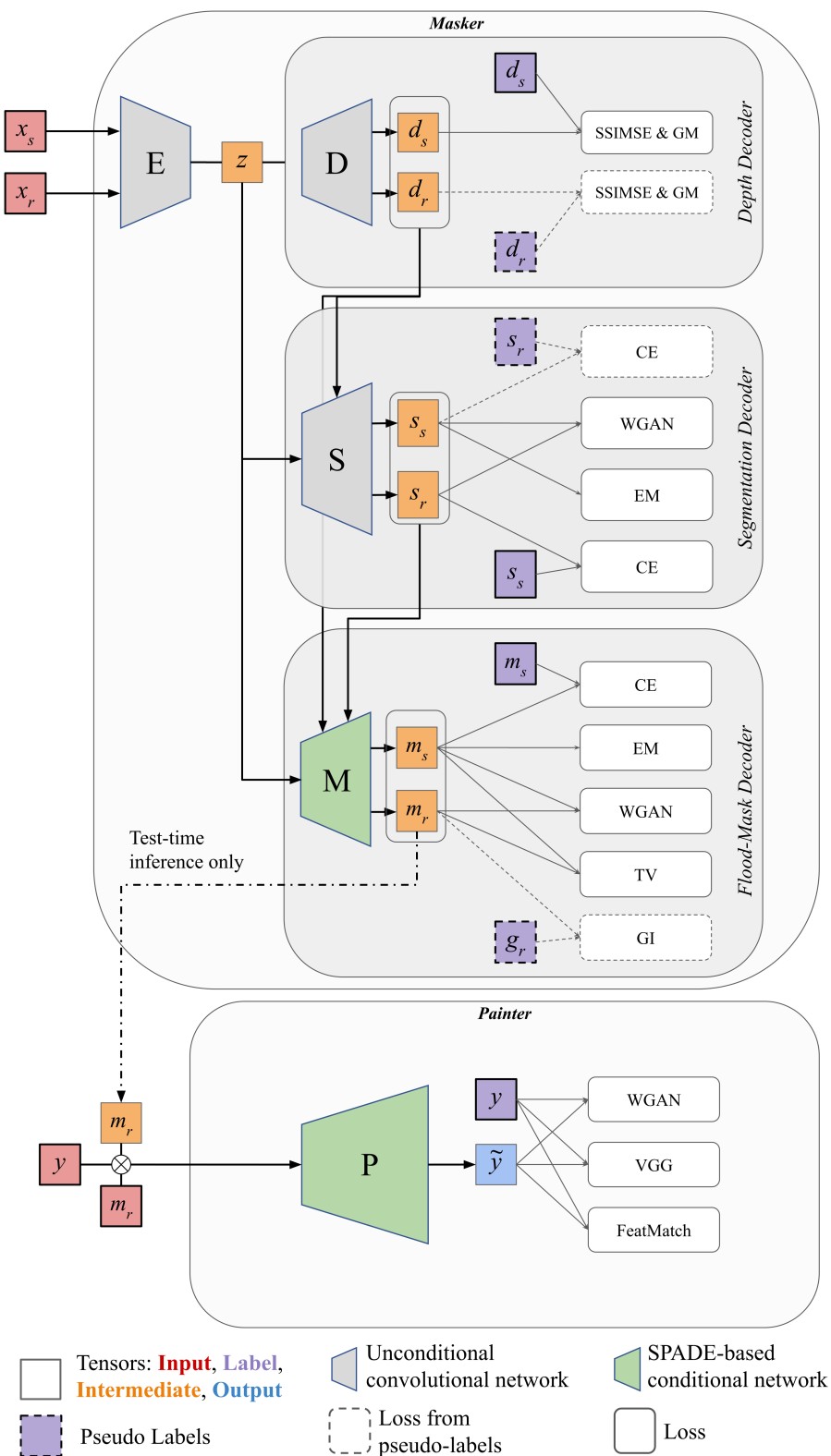

Figure 17: Detailed diagram of the training procedure of ClimateGAN. We highlight how the simulated and real data paths in the model are similar yet different. We use dashed arrows and boxes to emphasize that pseudo labels are only used as noisy signal in the beginning of training. All the losses represented in the white rounded boxes are detailed in Section 3.2 and Appendix B.

Input  Depth  Segmentation  Mask  Masked Input  Painted Input

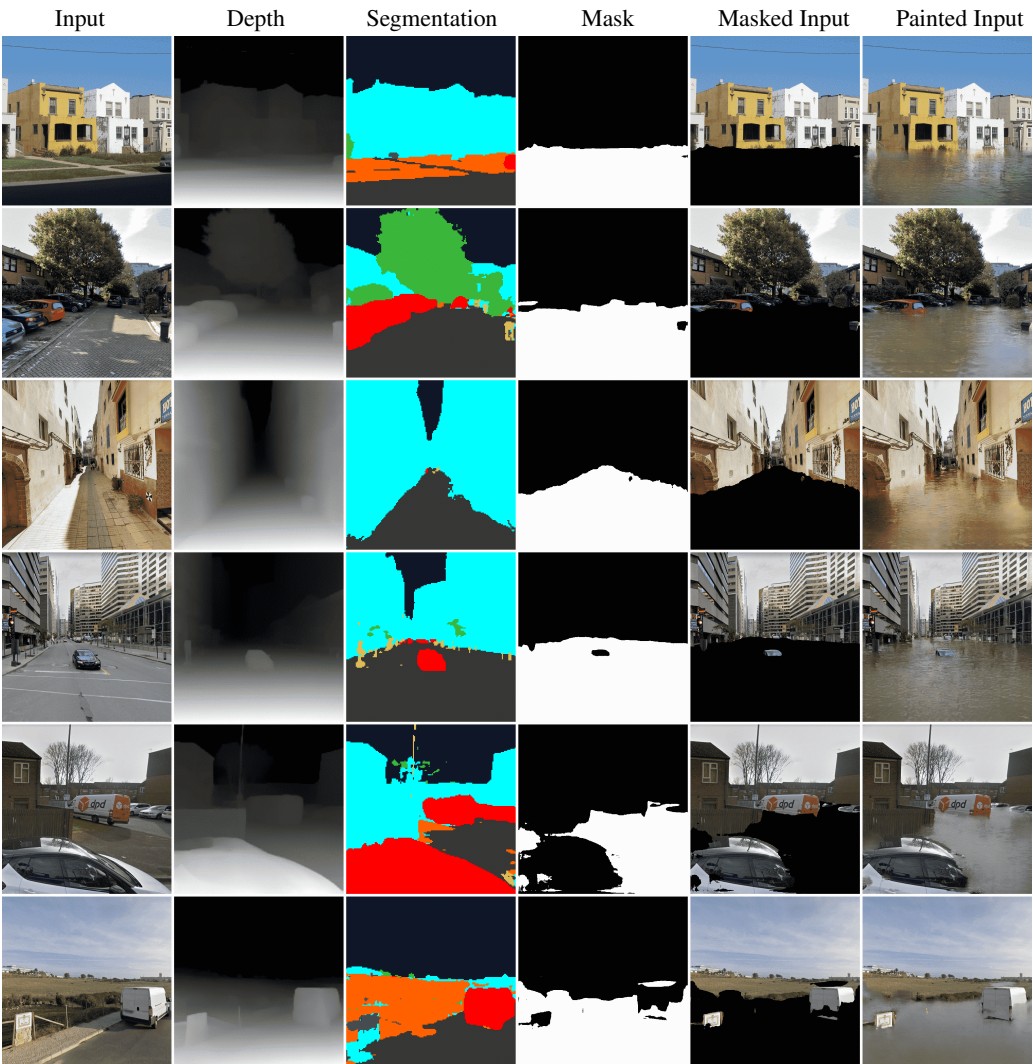

Figure 18: More samples of the full ClimateGAN forward pass: input image, inferences from the depth, segmentation and flood mask decoders of the Masker, followed by the masked input image fed to the Painter and finally the flooded image output by the Painter. Notably, the Painter is able to produce consistently contextualized water, with color and reflections relevant to the surrounding objects and to the sky. While generally able to appropriately understand a scene's perspective and circumvent objects, the Masker is however sometimes unable to predict plausible flood masks (as illustrated in the bottom two rows). It is difficult to exactly understand the source of this because both the depth and segmentation maps look generally appropriate.

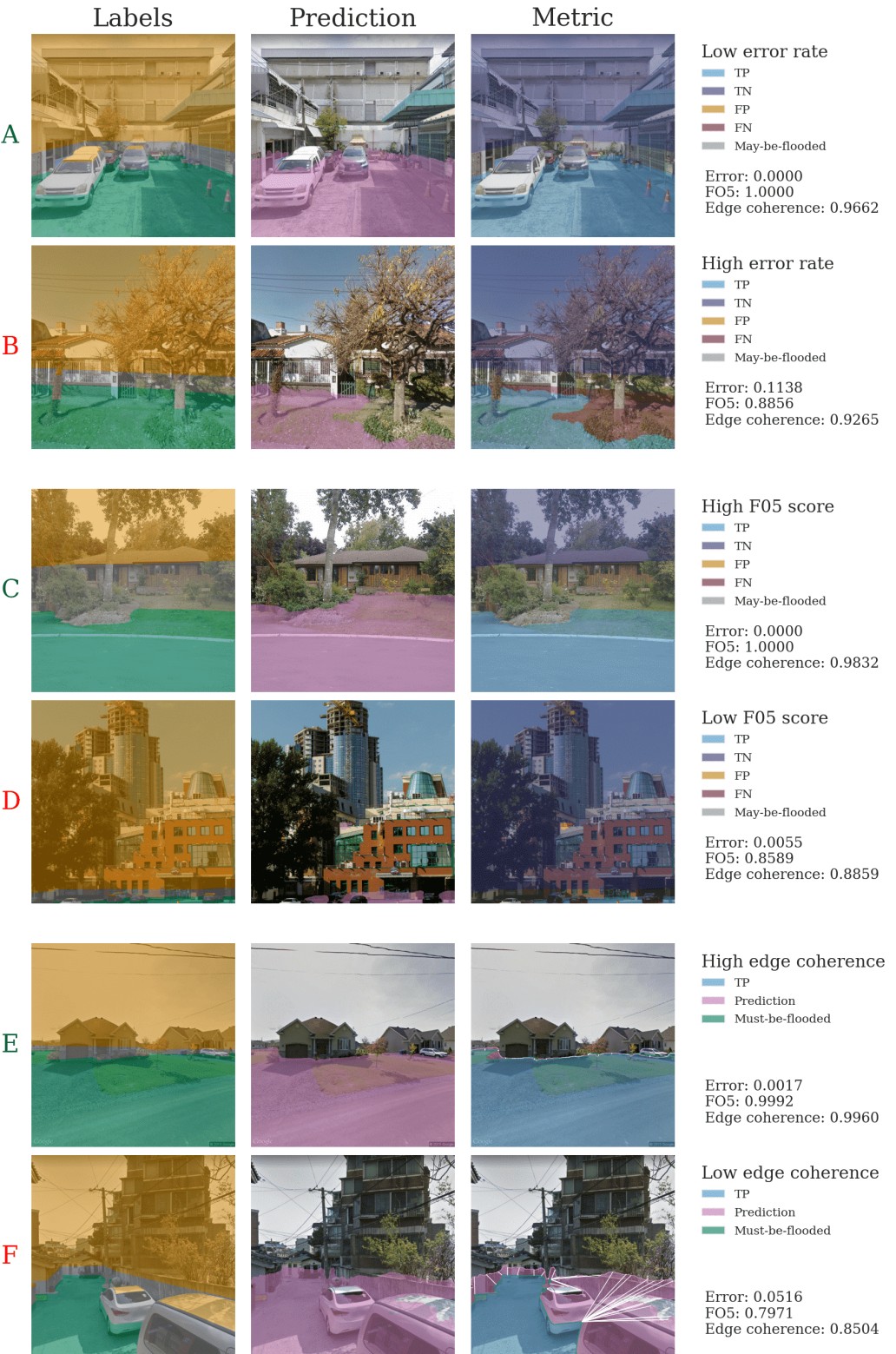

Figure 19: Examples of images that obtained good and bad—within the 2nd and 98th quantiles, respectively—Masker's predictions metrics. From top two rows to bottom two: error, F05 score and edge coherence. The first row of each metric corresponds to an image with good values of the metric. The white segments in the images illustrating the edge coherence indicate the shortest distance between the predicted and the ground truth mask. The legend of the column "Labels" is the same as in Fig. 12, and the images can be identified in Fig. 13 by the letters on the left.

