# OpenReview forum: "ClimateGAN: Raising Climate Change Awareness by Generating Images of Floods"
_ICLR.cc/2022/Conference — ICLR 2022 Poster_

### Official Review · Reviewer_TBDz · 2021-10-29

**Correctness:** 3
**Technical Novelty And Significance:** 2
**Empirical Novelty And Significance:** 2
**Recommendation:** 5
**Confidence:** 4

**Main Review:**

The ethical and societal motivation of the paper is clear and commendable.

I appreciate that the images generated from the virtual world with and without flooding are available, but I do not understand why the authors do not make available the real dataset. I understand that it contains images from Cityscapes and Mapillary, but it also contains new real images that the authors collected (as described in the “real data” section). I encourage the authors to make this other dataset available.

The main weakness of the paper is the limited technical contribution. The proposed ClimateGAN is given by combining two already existing models (DADA and GauGAN), with minor modifications. In section 3.4, the authors state that “A naive use of DADA for our task would miss a crucial difference” and “contribution of our Masker’s architecture is therefore to structure it as a multi-headed network”, but it is not clear to me if and how the authors have modified the architecture of DADA. By looking at Vu et al, 2019b, it seems to me that the architecture of the Masker is the same as DADA, apart from the last SPADE block “M” added after DADA.

I appreciate that the authors report in Figure 3 failure cases, but I it would be nice if the authors could comment more on such failures. Do they happen when the input image has a particular geometry or perspective? I would also include more failure cases in the paper (like the two in figure 18 in the appendix), where the Painter also fails at producing plausible flooding.

In the paper, the Masker and the Painter are trained independently and only combined at test-time. It would be interesting to see what happens if the Masker and the Painter are trained jointly and compare this to the results in the paper.

In section 4.1.2, the authors state “We considered a technique to be beneficial for the Masker if its inclusion reduced the error rate. In case of inconclusive error rate results, we considered an increase in the F05 score and finally in the edge coherence”. Why is this hierarchy considered for the three metrics? I think that an explanation is required.

The evaluation and results section are very dense in terms of information and their structure does not help the reader in grasping the key points. I would suggest to restructure them and to provide a bullet lists of the main takeaways/findings from the quantitative and qualitative analysis.


Human evaluation

Important details are missing about the set up of the human evaluation. The most important I would say is the number of evaluators. From the appendix, it seems that only three people took part in the evaluation. If this is the case, I believe that the results would not represent an accurate estimate of the performance comparison between the ClimateGAn and the other models.

About the “human evaluation” section it seems that ClimateGAN was compared against one model at a time. It would be interesting to compare all the considered models together and see how many times ClimateGAN produces the most realistic images among all models.

In the conclusion, the authors state that “we established, using human evaluations,  that our Painter alone is able to create realistic water textures when provided ground masks, and that its performance increases even further when provided with the Masker’s predictions”. However, I could not find such evaluation. From the paper, it seems that the human evaluation is for the Masker+Painter, not for the painter alone.


MINOR COMMENTS
-	Specify what the lambdas in the loss function represent.
-	In the “image-to-image translation” section, this sentence lacks the verb “CycleGAN (Zhu et al., 2017) this constraint. Also, what is the first category of IIT? Please clarify it in the text.
-	In the “conditional image synthesis”, I would add a reference where GauGAN is first mentioned.
-	I think it would be interesting to know what is the number of new real images that you were able to collect (apart from the ones taken from Cityscapes and Mapillary)
-	If my understanding is correct, the “comparables” section refers to other models used for comparison in the human evaluation section. I think it should be clearly stated in the section.




**Summary Of The Paper:**

In this paper, the authors introduce ClimateGAN, a framework for the generation of images of flooded scenarios in order to raise climate change awareness and prompt action. In particular, the authors consider the realistic case of scarcity of training data and propose and unsupervised approach. An evaluation of each ClimateGAN components is performed, both quantitative (through metrics) and qualitative (through human feedback). ClimateGAN is also compared against other generative modeling frameworks.


**Summary Of The Review:**

I believe that the purpose of the proposed model is topical and of interest. However, from a technical viewpoint, I believe that the contribution is not novel enough for acceptance at ICLR as ClimateGAN is given by combining two already existing models (DADA and GauGAN) with minor modifications.

---

> ### Author Response · Authors · 2021-11-17
> **First reply to reviewer TBDz (part 5/5)**
>
> ### Q10
>
> > In the conclusion, the authors state that “we established, using human evaluations, that our Painter alone is able to create realistic water textures when provided ground masks, and that its performance increases even further when provided with the Masker’s predictions”. However, I could not find such evaluation. From the paper, it seems that the human evaluation is for the Masker+Painter, not for the painter alone.
>
> We apologize if this was not clear in the manuscript, we will improve the wording. In Fig. 7, page 9, the last row shows how many times ClimateGAN (=Masker+Painter) was preferred to "Painted ground" which is the exact same Painter as ClimateGAN, but using ground masks as input (in addition to the context image). Those ground masks are obtained using a pre-trained, off-the-shelf and state-of-the-art HRNet segmentation model (Section 4.2), merging semantic labels (road, sidewalk, terrain, grass, ground) into a meta "ground" class.
>
>
> ---
>
> (Minor comments)
>
> ### Q11.1
> > Specify what the lambdas in the loss function represent.
>
> Thanks for pointing this out, we will clarify in the camera-ready version. The lambdas are hyperparameters, scalars that weigh the contribution of the various losses to the final objective to optimize.
>
> ### Q11.2
>
> > In the “image-to-image translation” section, this sentence lacks the verb “CycleGAN (Zhu et al., 2017) this constraint. Also, what is the first category of IIT? Please clarify it in the text.
>
> This is indeed a typo, the verb "relaxed" is missing: "(CycleGAN (Zhu et al., 2017) relaxed this constraint[...]"
>
> When it comes to "the first category of IIT", we mentioned it as "In the first case, initial IIT approaches relied on the existence of two aligned domains such as photographs and sketches of the same objects".
>
> ### Q11.3
>
> > In the “conditional image synthesis”, I would add a reference where GauGAN is first mentioned.
>
> Thanks for pointing it out. We will add it in the camera-ready version.
>
> ### Q11.4
>
> > I think it would be interesting to know what is the number of new real images that you were able to collect (apart from the ones taken from Cityscapes and Mapillary)
>
> In addition to existing data sets we collected a total of 1,894 images of non-flooded scenes:
>
> * 41 from user uploads to a website we created
> * 400 from stock picture websites
> * 1,453 from Google Street View in most census tracts across Canada to ensure diversity
>
> ### Q11.5
>
> > If my understanding is correct, the “comparables” section refers to other models used for comparison in the human evaluation section. I think it should be clearly stated in the section.
>
> It is true that the title to section 4.2 can be misinterpreted, would naming 4.2: "Comparables for human evaluation" alleviate the ambiguity in the reviewer's opinion?

---

> > ### Comment · Reviewer_TBDz · 2021-11-22
> > **Answer to authors' reply**
> >
> > First, I would like to thank the authors for their detailed answer and for clarifying many points. I think that many concerns I have raised have been addressed and the future version of the paper will be significantly improved. I understand the authors' point of view and I believe that the addressed problems is topical, but I remain of the opinion that the contributions and the novelty of the work are limited for this venue.
> >
> > With regards to the authors' question in Q6, I think that they should briefly explain in the main body of the paper the choice of metrics and the proposed hierarchy.
> > With regards to Q4, I would like to point out that my suggestion was to include more failure cases in the main body of the paper and to extend the discussion there, not in the appendix.
> >
> > Finally, I would like to point out that it is not clear in the paper (in its current form) that an important part of the contribution is the proposed evaluation strategy. I think that the contributions of the paper should be better highlighted (maybe a bulleted list might help).

---

> > > ### Author Response · Authors · 2021-11-22
> > > **Second reply to reviewer TBDz (part 1/1)**
> > >
> > > **Thank you** very much for replying and commenting a second time. We are happy to read that we did address your concerns. The remaining challenge around novelty is part of the discussion and we would like to take a moment to shed a different light to the conversation.
> > >
> > > ICLR clearly states in its [call for papers](https://iclr.cc/Conferences/2022/CallForPapers) that relevant subject areas for the conference includes "applications in vision, [...], sustainability, [...]". We believe our paper to be in-scope with those targets as we introduce an *application* of modern *computer vision* techniques (SPADE, DADA etc.) to help *climate change* mitigation. In this context, we also would like to point out that the expected "novelty" of applications papers is different than more fundamental, theoretical or general purpose papers and contributions in areas like "reinforcement learning", "compositional modeling" etc.
> > >
> > > We sincerely believe that novelty must be interpreted differently depending on a paper's category. In other words, pushing for architectural innovation for the sole sake of "technical novelty" is dangerous for our field, while said technical novelty can lie in so many other features of a paper, especially for applied research (like the task, the data set and the evaluation procedure in our case). This is at the core of the concept of an "application" and we know for a fact that the intention of those explicit mentions by ICLR chairs is *meant* to encourage such research.
> > >
> > > ---
> > >
> > > * **Q6**
> > >     * We agree with your suggestion and will do so
> > > * **Q4**
> > >     * This would certainly improve the paper and we will do our best the include this without detriment of other important content, within page constraints. Suggestions for how to make space would be very welcome.
> > > * **Evaluation strategy & contributions**
> > >     * We are sorry the reviewer felt this way, as we were careful to include it, along with an explicit list of contributions, in both the introduction and the conclusion. Here is the relevant text from the manuscript:
> > >         * Introduction: "Our contributions are: proposing and motivating the novel task of street-level flood generation, a data set of pairs of images with/without flooding from a virtual world, the ClimateGAN model which includes a novel multitask architecture for generating geometry- and semantics-informed binary masks and *a procedure to thoroughly evaluate it in the absence of ground-truth data.*"
> > >         * Conclusion: "Our contributions comprise a data set of images and labels from a 3D virtual world, a novel architecture ClimateGAN for the task of street-level flood generation, and *a thorough evaluation method to measure the performance of our system*"
> > >             * We will make our best to try and isolate those as a bullet list, under the page limit constraint
> > >     * Would a third mention in the introductory paragraph of Section 4 help make our case for the evaluation strategy in the reviewer's opinion?

---

> > > > ### Comment · Reviewer_TBDz · 2021-11-24
> > > > **Second answer to authors' reply**
> > > >
> > > > I would like to thank the authors again for their thorough reply and interesting discussion.
> > > >
> > > > With regards to authors' question, yes I think that a third mention at the beginning of Section 4 could be helpful.
> > > >
> > > > With regards to the call for papers, I never said that the topic of the paper is out of scope. My argument is about the novelty of the contribution. Anyway, given the discussion with the authors and the different point of view they presented, I have raised my score accordingly.

---

> ### Author Response · Authors · 2021-11-17
> **First reply to reviewer TBDz (part 4/5)**
>
> ### Q7
>
> > The evaluation and results section are very dense [...]. I would suggest to restructure them and to provide a bullet lists of the main takeaways/findings from the quantitative and qualitative analysis.
>
> We appreciate this piece of feedback and acknowledge that there is room for improvement in organizing the presentation of evaluation methods and results. Would it be more readable if we provided the results right after the introduction of the method, instead of splitting these into two sections? We also think that we could rename the subsections to more explicitly reflect the content.
>
> We particularly welcome the suggestion of a bulleted list to summarize the findings and takeaways. We propose the following:
>
> * Of the 6 techniques proposed to improve the Masker---pseudo labels, a depth estimation head, a segmentation head, SPADE, DADA for segmentation and DADA for the Masker---5 of them (all except DADA for the Masker) positively and significantly contributed to the performance, according on our proposed metrics. These results were obtained through an ablation study, for which we obtained confidence intervals using the bootstrap. This is shown in Figure 5.
> * The best-performing models were those which included all or most of the techniques identified as helpful in the ablation study. This is shown in Figure 6.
> * Our proposed architecture for the Masker largely outperformed the two baselines included for comparison (ground segmentation via HR-Net and InstaGAN). This is shown in Figure 6.
> * The outputs of ClimateGAN (_flooded_ images) were consistently selected to be more realistic and look more like an actual flood---by a total of 147 human evaluators---than the images of 5 alternative models. This is shown in Figure 7.
>
> ---
> ### Q8
> > Important details are missing about the set up of the human evaluation. The most important I would say is the number of evaluators. From the appendix, it seems that only three people took part in the evaluation. [...]
>
> We believe there must be a misunderstanding about this detail. In fact, 147 evaluators took part in the study (done via the Mechanical Turk platform) and we gathered 2700 total evaluations, 3 for each _pair_ of images. For each image generated by ClimateGAN, we compared it to 5 other approaches (MUNIT, InstaGAN, CycleGAN, InstaGAN with Mask and Painted Ground), and each one of these pairs was seen by 3 evaluators. We will further clarify this in the paper.
>
> ---
> ### Q9
> > [I]t seems that ClimateGAN was compared against one model at a time. It would be interesting to compare all the considered models together and see how many times ClimateGAN produces the most realistic images among all models.
>
>
> We agree with the reviewer that ideally it would be interesting to compare all models against all. However, this approach would introduce methodological complications, both in the statistical tools at hand and most especially because of the biases known to affect human evaluators when performing multi-way comparisons. For instance, laterality has been well studied to largely impact visual inspection (for example, Ossandón et al., 2014) and two-alternative forced choices is known to be an easier and more reliable task for human evaluators, reason why it is the common approach to measure sensitivity to stimuli or preferences in the cognitive sciences (Link and Heath, 1975). In addition, recent research in GAN evaluations also recommends making pairwise comparisons in order to measure user preference (Zhou et al., 2019). We contacted the authors but because their system is not currently operational we had to implement the evaluation ourselves on Mechanical Turk.
>
> Furthermore, in our case we are mainly interested in knowing whether the images produced by our model are preferred over other alternatives, rather than comparing these alternatives against each other. Therefore, we believe that comparing ClimateGAN against four other models individually should allows to conclude whether our model is preferred over the rest or not. According to the results, ClimateGAN images were preferred in the vast majority of pairwise trials.
>
> **References**
>
> * Ossandón, J. P., Onat, S., & König, P. (2014). Spatial biases in viewing behavior. Journal of Vision
> * Link, S. W., & Heath, R. A. (1975). A sequential theory of psychological discrimination. Psychometrika
> * Zhou, S, Gordon, M, Krishna, R, Narcomey, A, Fei-Fei, L & Bernstein, M (2019). HYPE: A Benchmark for Human eYe Perceptual Evaluation of Generative Models

---

> ### Author Response · Authors · 2021-11-17
> **First reply to reviewer TBDz (part 3/5)**
>
> ### Q6
>
> > In section 4.1.2, the authors state “We considered a technique to be beneficial for the Masker if its inclusion reduced the error rate. In case of inconclusive error rate results, we considered an increase in the $F_{\beta=0.5}$ score and finally in the edge coherence”. Why is this hierarchy considered for the three metrics? I think that an explanation is required.
>
> We agree that this is an important aspect of the evaluation, but due to space limitations we moved it to the appendix. Further details about the rationale behind each metric and the hierarchy can be found in Appendix C.2 Metrics. We extend the discussion here, for completeness.
>
> The development and selection of the three metrics that we concluded are the most indicative of the quality of the masks was based on an extensive empirical (visual) evaluation of output images from multiple models, with the values of several metrics and prediction masks such as the ones provided in Figure 19 in the Appendix. We believe that the selection of these metrics and the proposed hierarchy, based on our systematic evaluation, is one contribution of the paper.
>
> We concluded that the most decisive aspect in the perceptual quality of the masks, and hence the final flooded image, is the number of incorrectly classified pixels with respect to the size of the images. That is, large areas incorrectly predicted as water (false positives) and large areas that should be _flooded_ incorrectly missed (false negatives) are very likely to be perceived as unrealistic. The error rate captures precisely this and therefore we selected it as the most important metric.
>
> However, the error rate fails to capture _all cases_ where the output of the mask may be perceived as incorrect. For example, images with a relatively small area to be _flooded_ are unlikely to yield outputs with a large mask, as compared to the image size. However, if the mask were completely missed, the output image would appear incorrect, whereas the error rate would be small (see image D in Figure 19). In order to take into account the largely varying size of the true masks and its impact on the perceptual correctness, we proposed to also take into account the $F_{\beta=0.5}$ score. We found false positives (incorrectly _flooded_ areas) to have a larger impact on our perceptual evaluation than false negatives, reason why we decided to weigh more precision than recall.
>
> Finally, we noted that neither the error rate nor the $F_{\beta=0.5}$ score take into account the geometry of the predictions, but just the number of errors. In order to characterize the correctness of the edge of the predicted masks, we proposed the _edge coherence_ metric, which measures the _parallelness_ between predicted mask and ground truth. However, we noted that this metric in and on itself is not sufficient to characterize the the quality of the predictions of one model on a set of images, but it rather complements the other two metrics. In other words, error rate and $F_{\beta=0.5}$ score being similar, the edge coherence may allow to further discriminate the quality of the predictions.
>
> We hope this sheds some light on the rationale behind the choice of metrics and the proposed hierarchy. We kindly ask the reviewer whether they think we should include part of this discussion in the main body of the paper, rather than in the appendix.

---

> ### Author Response · Authors · 2021-11-17
> **First reply to reviewer TBDz (part 2/5)**
>
> ### Q4
>
> > [...] it would be nice if the authors could comment more on such failures. Do they happen when the input image has a particular geometry or perspective? I would also include more failure cases in the paper (like the two in figure 18 in the appendix), where the Painter also fails at producing plausible flooding.
>
> Regarding failure cases of the model, we are willing to include more images in the appendix and extend the discussion to shed more light. Some of our conclusions are the following:
>
> * First, the Painter seems to systematically generate realistic texture of water, successfully incorporating the context of the scene such as reflections of buildings on the water, _provided the mask is reasonably accurate_. In other words, we have not identified failure patterns of the Painter, specifically.
> * The output of the Painter often mitigates small failures in the accuracy of the mask. In other words, even in some cases where the mask is not perfect, the output image of the Painter may look perceptually realistic.
> * The source of failure cases, where the final image does not look perceptually realistic, is instead the Masker, given the more challenging nature of the task---predicting a mask for a hypothetical flood, taking into account the geometry of the scene, semantics, etc. The following are the kinds of images where we have found the Masker to perform sub-optimally at times:
>     * Images where the sky or the ground are not visible
>     * Images with a large object, such as a car or a truck, taking a large fraction of the foreground.
>     * Wide open views of grass fields - the edges of the water mask are sometimes incorrect.
>     * Scenes with multiple people in the scene - the edges of the water mask around all objects are sometimes inaccurate, although the perceptual error is mitigated by the Painter.
>     * Scenes with much vegetation, such as bushes
>     * Urban scenes with a pronounced slope. Surprisingly, the _painted_ image sometimes looks realistic in these cases, despite an inaccurate Mask.
>
> We hypothesize that the reason for lower accuracy of the Masker in these cases is the lack of such images in the training set, beside the additional difficulty intrinsic to some of these images, even to a human annotator.
>
> We will include more of such images in the appendix, and extend the discussion, based on the items mentioned above.
>
> ---
> ### Q5
>
> > In the paper, the Masker and the Painter are trained independently and only combined at test-time. It would be interesting to see what happens if the Masker and the Painter are trained jointly and compare this to the results in the paper.
>
>
>
> This is indeed a great topic for future research, which we have actually studied, but we could not touch upon in the current manuscript due to page limitations. One major constraint to train the Painter and the Masker jointly is that two of the Painter's losses (Feature Matching Loss and Perceptual Loss) require the input data and generated data to be compared, and should therefore come from the same "category": flooded scenes. In other words, both those losses compare the input and the output to encourage the Painter to produce more realistic water. On the other hand, the Masker never processes such data: it produces masks from _non-flooded_ images.
>
> Given this, we have tried the 2 following approaches:
>
> 1. Remove the two aforementioned losses from the Painter's training procedure, and train it jointly with the masker from non-flooded data (a variation of Eq. $(7)$):
>
>     $$
>     \tilde y(x) = P(\epsilon, (1 - Masker(x)) \odot x) \odot Masker(x) + x \odot (1 - Masker(x)) \hspace{1cm} (1)
>     $$
>
>       This procedure (as expected) not only makes training the masker more difficult but we could not make the Painter to produce realistic water *at all*
>
> 2. Keep the two losses, train the Painter and Masker from different data sources (flooded/non-flooded) but use the loss from the Discriminator of the Painter to train the Masker. In other words, train $P$ and $M$ as described in the paper but add a loss ($P$ is frozen here):
>
>     $$
>     L_{GAN_P}(D_P(\tilde y(x)))
>     $$
>
>       where $L_{GAN_P}$ is a GAN loss (cross-entropy or $L_2$ for instance), and $\tilde y(x)$ is defined in $(1)$ above. Since the Discriminator $D_P$ in the Painter's adversarial training procedure is supposed to assess how "realistic" its output is, then this takes into account not only *how* the water is painted, but also *where* it is. Unfortunately this proved to be very challenging in terms of computational complexity, memory footprint and most of all, convergence. We could not find a proper scheduling/scaling mechanism for $L_{GAN_P}$ to be informative enough to improve the Masker's performance.
>
> TL;DR we definitely agree with the reviewer that this is an interesting idea to pursue further, yet so far our attempts have failed to *improve* on the independent training scheme.

---

> ### Author Response · Authors · 2021-11-17
> **First reply to reviewer TBDz (part 1/5)**
>
> We are grateful the reviewer took the time to express detailed concerns and questions, which we hope will lead to a fruitful discussion and potentially clarify our methods and contributions. We are glad they agree on the relevance and interest of our research, we hope we can convince them that we not only bring some novelty in the Masker's architecture (which we acknowledge is incremental), but that our novelty contributions lie well beyond that: in the task itself, the data set and the evaluation procedure.
>
> Since the reviewer's feedback regarding novelty is shared by other reviewers, we wrote a summary response as a new comment to this OpenReview Forum. We hope they will find it useful.
>
>
> ---
> ### Q1
> > I do not understand why the authors do not make available the real dataset. [...] I encourage the authors to make this other dataset available.
>
> The reason why we have not made the real data set available is because we are not allowed due to license issues. However, we agree that facilitating the recreation of the data set is important for future work and reproducibility. Therefore, we will release the IDs of Mapillary images used and release the URLs of images we got from search engines. We will also release guidelines about making this data set.
>
> ---
> ### Q2
> > The main weakness of the paper is the limited technical contribution. The proposed ClimateGAN is given by combining two already existing models (DADA and GauGAN), with minor modifications. [...]
>
> This concern about contributions is shared with reviewer `PcPf` (Q2) and we adapt our reply here.
>
>
> The reviewer is right that the *two-stage design* is an important part of our contribution. We however believe our main contribution to lie in the Masker model, since we do not claim contributions for our GauGAN Painter. We would like to emphasize that most of our contributions are introduced to make the latter as robust as possible in the real world, facing actual human users:
>
> 1. a virtual world from which we collect a data set, tailored to our task and which we open-sourced to share with the research  community at large. This data set comprises diverse scenes from urban, sub-urban and rural areas. Moreover, in addition to pairs of image with or without water, we include labels: metric depth and segmentation. This new data set contains more than 20 000 multi-modal samples.
>
> 2. an architecture that combines processing blocks in a new way, for a new  purpose. In other words, we do not simply reuse DADA or SPADE: we combine them in a way that makes pixel-level (input), geometric (depth) and semantic (segmentation map) available *together*, for the Masker to produce well contextualized predictions. This is an entirely new combination because we have a very unique task at hand.
>     1. We would like to respectfully point out to the reviewer that our goal in this application is not to create a model fit for other tasks than the one we propose.
>     2. In the same spirit, novelty in AI should not, in our opinion, be restricted to new "blocks" or "operations" never seen before. Leveraging already existing methods for new purposes, in new combinations, proving that they work in other settings than those envisioned by their inventors is a critical part of science.
>
>   3. an evaluation procedure to carefully assess the contribution of each component. As emphasized in the paper, the complexity of our task is driven by the fact that the Masker predicts not what *is* in the original image, but rather what *should be*, without any ground truth available. To add to this difficulty, there are multiple acceptable solutions to our problem and we had to come-up with our own, tailored evaluation procedure to be able to capture this. An important part of our contribution is therefore the design of a fair, systematic and robust evaluation strategy, absent any pre-existing standard procedure.
>
> ---
>
> ### Q3
>
> > it is not clear to me if and how the authors have modified the architecture of DADA. By looking at Vu et al, 2019b, it seems to me that the architecture of the Masker is the same as DADA, apart from the last SPADE block “M” added after DADA.
>
> As the reviewer rightfully points out, DADA is *contained* in our Masker. The novelty of the Masker is to re-purpose SPADE blocks in order to produce *binary masks* conditioned on multi-modal information: a shared latent vector, depth and segmentation.
>
> Taking a step back, DADA builds on ADVENT by informing segmentation with depth and sharing representations to do so. As explained, the Masker's task is not to predict what *is* in the image, rather what *should be* there. It is not a segmentation task. But we believed segmentation should be useful for this. The question we therefore tried to answer was: how can we build on DADA to inform the Masker with *both* segmentation and depth? This is where we introduce this novel use of SPADE conditioning blocks, trained jointly with the segmentation and depth heads with a shared encoder network.

---

> ### Author Response · Authors · 2021-11-19
> **Following up rebuttal**
>
> Dear reviewer,
>
> We know the timeline for the discussion period is tight and we wrote rather lengthy responses to address all aspects of your review. We would love to use this chance to discuss with you any concerns that may remain open. We would appreciate it if you could confirm that you have read our responses and let us know whether we have successfully address the concerns or there are further points to discuss.
>
> Thanks for your time invested in this review!

---

### Official Review · Reviewer_PcPf · 2021-10-31

**Correctness:** 3
**Technical Novelty And Significance:** 2
**Empirical Novelty And Significance:** 2
**Recommendation:** 5
**Confidence:** 4

**Main Review:**

Strengths:
 - This is a good application by leveraging the cutting-edge GAN-related techniques on an interesting area of generating flooded images.
 - The two-stage design with a Masker and a Painter is useful for synthesizing floods on images.

Weaknesses:
 - The motivation of this work is described confusingly. First, the meaning of generating the flooded images is not well explained. Second, generating flooded images is not well related to the climate change. Thus, this work seems not that meaningful for climate change.
 - The main contribution of proposed ClimateGAN is the two-stage design for synthesizing floods on images, but each stage (Masker/Painter)  is designed with limited novelty based on some popular models and losses (refer to the ablation study in Section 4.1.2), making the whole work is not strong enough.
 - The comparison is not convincing. First, to evaluate the whole ClimateGAN, it's better to consider the state-of-the-art image-to-image translation methods, such as DRIT++, GauGAN, TSIT, DUNIT, etc. Second, to evaluate the Masker for flooded mask estimation, it's better to consider the state-of-the-art methods on both image-to-image translation and image segmentation (or maybe other related techniques). Besides, the data for training and testing should also be well considered for fair comparison.
 - If the two-stage design is the most important concept of this work, this should be well demonstrated with more insightful analysis theoretically and empirically.

**Summary Of The Paper:**

This paper proposes to generate images of floods, that is, simulate photo-realistic floods, with a model named ClimateGAN, which consists of a Masker to generate masks for floods and a Painter to draw floods on images. The proposed ClimateGAN is a two-stage solution for generating flood masks on input images and painting the images with floods respectively, and each stage (Masker and Painter) is designed based on many popular related techniques, like DADA for depth-aware semantic segmentation, SPADE for injecting extra information into cGANs, and WGAN as well as TV, BCE and EM losses. For training ClimateGAN, this work also collects a real dataset with street-level floods and generates a simulated dataset using Unity3D. The outputs of Masker and Painter are evaluated separately compared to several previous image-to-image translation methods, also the ablation study is conducted to validate the effectiveness of each component in ClimateGAN.

**Summary Of The Review:**

The paper can be regarded as a particular application for synthesizing flooded images with street-level cityscape images, however, since this work is not well motivated, yet the contribution and comparison are not strong enough, overall, I vote for rejecting.

---

> ### Author Response · Authors · 2021-11-17
> **First reply to reviewer PcPf (part 3/3)**
>
> ### Q3
>
> > The comparison is not convincing. First, to evaluate the whole ClimateGAN, it's better to consider the state-of-the-art image-to-image translation methods, such as DRIT++, GauGAN, TSIT, DUNIT, etc.
>
>
> We are not entirely sure what the reviewer means by considering IIT methods to evaluate the whole ClimateGAN. In this paragraph, we answer what we *understand the question to be*, and kindly ask the reviewer to clarify their question's intention if it is different.
>
> If they mean translation from non-flooded to flooded domain directly, that could be performed with methods disentangling style and content, we compared our model with MUNIT. We did not pursue this direction and compare with models such as TSIT, given the unsatisfactory results obtained with MUNIT, that we attribute to the nature of our task : the "style" in our case should not be applied on the whole image, as we only want to alter the part that could be flooded.
>
> Besides, if they mean producing masks without supervised cross entropy but rather a GAN loss, we do not know of any evidence in the literature indicating  IIT is superior to domain adaptation with supervised cross entropy. Moreover, we actually compared our model with InstaGAN which does IIT for masks as a sub-task and found our model was preferred over InstaGAN. One limitation to using SOTA methods that require segmentation masks is that there are, generally, no masks available in the real world, therefore both IIT and domain adaptation are needed, which is what our model does.
>
> We thank the reviewer for pointing out methods relying on instance-awareness such as DUNIT. However, they are not applicable to our case. Indeed, DUNIT uses instance-awareness, to make sure that the generated image still includes the certain objects. For example, a car on the street will be here no matter it is daytime or night. However, this setting does fit our use-case because the flooded areas will typically include multiple instances which can be *partially* and *differently* transformed (*i.e.* hidden under water). For instance, the ground may not be entirely flooded if there is a slope or the top half of a car may be above the water level. Finally, the area to be flooded is not any specific object that already exists in the original image. In this context, the concept of "instance" to transform is ill-defined in the context of our specific task.
>
> GauGAN does not apply directly to our case either since it relies on a segmentation map, and obtaining the precise locations of where water should be generated is *precisely* the role of our Masker. We did however use this architecture for the Painter for which we do not claim any contribution in terms of the architecture of the network.
>
> Besides, the nature of our task is such that high diversity in the outputs is not a criterion for the selection of the model, therefore we did not pursue exploration of methods that were particularly good at generating diverse outputs such as DRIT++.
>
> ---
>
> ### Q4
>
> > Second, to evaluate the Masker for flooded mask estimation, it's better to consider the state-of-the-art methods on both image-to-image translation and image segmentation (or maybe other related techniques).
>
> Regarding the evaluation of the flooded mask estimation, we do compare with ground segmentation obtained with a SOTA model, HRNet trained on the Cityscapes data set.

---

> ### Author Response · Authors · 2021-11-17
> **First reply to reviewer PcPf (part 2/3)**
>
> ### Q2
>
> > The main contribution of proposed ClimateGAN is the two-stage design for synthesizing floods on images, but each stage (Masker/Painter) is designed with limited novelty based on some popular models and losses (refer to the ablation study in Section 4.1.2), making the whole work is not strong enough.
>
> &
>
> > If the two-stage design is the most important concept of this work, this should be well demonstrated with more insightful analysis theoretically and empirically.
>
> The reviewer is right that the *two-stage design* is an important part of our contribution. It is not, however, the "most important concept", which we believe to lie in the Masker model. We would like to emphasize that most of our contributions are introduced to make the latter as robust as possible in the real world, facing actual human users:
>
> 1. a virtual world from which we collect a data set, tailored to our task and which we open-sourced to share with the research  community at large. This data set comprises diverse scenes from urban, sub-urban and rural areas. Moreover, in addition to pairs of images with or without water, we include labels: metric depth and segmentation. This new data set contains more than 20 000 multi-modal samples.
>
> 2. an architecture that combines processing blocks in a new way, for a new  purpose. In other words, we do not simply reuse DADA or SPADE: we combine them in a way that makes pixel-level (input), geometric (depth) and semantic (segmentation map) available *together*, for the Masker to produce well contextualized predictions. This is an entirely new combination because we have a very unique task at hand.
>     1. We would like to also note that our goal in this application is not to create a model fit for other tasks than the one we propose.
>     2. In the same spirit, novelty in AI should not, in our opinion, be restricted to new "blocks" or "operations" never seen before. Leveraging already existing methods for new purposes, in new combinations, proving that they work in other settings than those envisioned by their inventors is a critical part of science.
>
>   3. an evaluation procedure to carefully assess the contribution of each component. As emphasized in the paper, the complexity of our task is driven by the fact that the Masker predicts not what *is* in the original image, but rather what *should be*, without any ground truth available. To add to this difficulty, there are multiple acceptable solutions to our problem and we had to come-up with our own, tailored evaluation procedure to be able to capture this. An important part of our contribution is therefore the design of a fair, systematic and robust evaluation strategy, absent any pre-existing standard procedure.
>
>
> Regarding the specific question of evaluating the two-stage design, we believe those elements to demonstrate the performance of our approach:
>
> 1. The Masker's performance, along with the contribution of many of its components is evaluated in detail in Sections 4.1 and 5.1, demonstrating nearly every single addition to its structure improved performance over both baselines and ablated Maskers (more in Figure 15). Using the Bootstrap method we provide statistical significance with 99 % confidence, demonstrating the superiority of the masker and the usefulness of its components
> 2. The visual quality of the two-stage approach is measured in Section 5.1 against: baseline IIT GANs (CycleGAN and MUNIT), a mask-based IIT GAN (InstaGAN) and an isolated Painter using masks from a pre-trained segmentation model. ClimateGAN is judged to be superior well beyond confidence intervals by a pool of almost 150 human evaluators who took part in our comparison experiments.
>
> Note that we do not claim contributions to the Painter's architecture and use an adapted version of GauGAN. We find that combining the Masker and the Painter is, indeed, the most visually appealing approach. In addition, results from section 5.1 show that it is also the most robust considering the wide variety of scenes in our test set, which we constructed to include visuals of different kinds of scenes and other countries than the training set (4.1).

---

> ### Author Response · Authors · 2021-11-17
> **First reply to reviewer PcPf (part 1/3)**
>
> We thank the reviewer for sharing their questions, ideas and concerns with us. We hope to clarify our motivation here as well as provide insights regarding how we selected the comparable approaches and a different perspective on our contributions.
>
> Please note that some concerns on novelty are shared with reviewer `TBDz` and that we wrote a summary response for all reviewers as a new comment in this OpenReview Forum.
>
> ---
> ### Q1
> > The motivation of this work is described confusingly. First, the meaning of generating the flooded images is not well explained. Second, generating flooded images is not well related to the climate change. Thus, this work seems not that meaningful for climate change.
>
>
> The motivation behind our work is to reduce the psychological distance that people have with regards to climate change by generating personalized images of potential impacts of extreme weather events.
>
> The "personal" aspect comes from the fact that a user will be able to query an image from Google Street View of any location they relate to: their home, their parents', children's, or friends', but also familiar places like parcs, schools or even local/regional/national/global landmarks.
>
> The meaning of "generating floods" in this context is to create a modification of said Street View image, as if it were being flooded. As explained in our paper, this translates into creating a water "plane" around 1m above the floor---which is a realistic expected water level for climate change-related flooding events [1].
>
>
> In fact, flooding is getting increasingly heavier and more frequent due to the effects of climate change [1, 2, 3], yet most people feel that they are not personally impacted by it, and so fail to take climate action [4].
>
>
> In this context, research has shown that the climate narrative should be made present, local, personal and visual [4, 5, 6].  Such image-centric approaches have focused for instance on selecting relevant photographs to represent the extent of climate change impacts [7, 8] as well as on using artistic renderings of possible future landscapes [9] and even immersive video games[10] and virtual reality experiences [11].
>
> There is however, to our knowledge no tool that is both available to the general public and able to create systematic, robust and realistic renderings of *any* location using AI, which motivates our research in creating such a model.
>
>
> * [1] Vousdoukas et al. [Global probabilistic projections of extreme sea levels show intensification of coastal flood hazard](https://www.nature.com/articles/s41467-018-04692-w). Nature Communications. 2018
> * [2] Kulp et al. [New elevation data triple estimates of global vulnerability to sea-level rise and coastal flooding](https://www.nature.com/articles/s41467-018-04692-w). Nature Communications. 2019
> * [3] Brunner et al. [An extremeness threshold determines the regional response of floods to changes in rainfall extremes](https://doi.org/10.1038/s43247-021-00248-x). Commun Earth Environ. 2021
> * [4] van der Linden et al. [Improving Public Engagement With Climate Change: Five "Best Practice" Insights From Psychological Science](10.1177/1745691615598516). Perspect Psychol Sci. 2015
> * [5] Chapman et al. [Climate visuals: A mixed methods investigation of public perceptions of climate images in three countries](https://www.sciencedirect.com/science/article/abs/pii/S095937801630351X) Global Environ. Change. 201.
> * [6] Wang et al. [Public engagement with climate imagery in a changing digital landscape](https://wires.onlinelibrary.wiley.com/doi/abs/10.1002/wcc.509) Wiley Interdisciplinary Rev. Climate Change. 2018
> * [7] Corner et al. [Talking Climate: From Research to Practice in Public Engagement](https://link.springer.com/book/10.1007/978-3-319-46744-3). Springer, 2016.
> * [8] Sheppard. [Visualizing Climate Change: A Guide to Visual Communication of Climate Change and Developing Local Solutions](https://www.emerald.com/insight/content/doi/10.1108/meq.2012.08323eaa.012/full/html). Management of Environmental Quality. 2012
> * [9] Giannachi. [Representing, performing and mitigating climate change in contemporary art practice](https://direct.mit.edu/leon/article-abstract/45/2/124/97857/Representing-Performing-and-Mitigating-Climate?redirectedFrom=fulltext) Leonardo. 2012
> * [10] Angel et al. [Future delta 2.0 an experiential learning context for a serious game about local climate change.](https://dl.acm.org/doi/10.1145/2818498.2818512) Association for Computing Machinery. 2015
> * [11] Ahn, Sun Joo. [Embodied experiences in immersive virtual environments: Effects on pro-environmental attitude and behavior.](https://stanfordvr.com/mm/2011/ahn-embodied-experiences.pdf) Stanford University, 2011.

---

> ### Author Response · Authors · 2021-11-19
> **Following up rebuttal**
>
> Dear reviewer,
>
> We know the timeline for the discussion period is tight and we wrote rather lengthy responses to address all aspects of your review. We would love to use this chance to discuss with you any concerns that may remain open. We would appreciate it if you could confirm that you have read our responses and let us know whether we have successfully address the concerns or there are further points to discuss.
>
> Thanks for your time invested in this review!

---

> > ### Comment · Reviewer_PcPf · 2021-11-25
> > **Reply to authors' rebuttal**
> >
> > I'd like to thank the authors for their detailed reply about my concerns.
> >
> > 1. I consider the description about the motivation to be better now, where the expression is more accurate. Essentially, this study is far away from the climate change but is close to provide help on the mentioned "psychological distancing" for people about flood by producing physically plausible and visually realistic flooded images (especially of the familiar places like home), from this point of view, FloodGAN might be more accurate than ClimateGAN (this is not only related to the name itself but the core concept of the writing).
> > 2. I agree with that the important applications about AI on very valuable fields, especially related to our life like climate change and flood cognition, is acceptable and meaningful, however, this should also be insightful enough. Actually, the motivation (revised) is good and I think that the contribution of this work might not be technical but interdisciplinary, however, the experiments are designed technically without any consideration on the related "psychological distancing", also the method is written technically without insights of interdisciplinarity.
> >
> > Based on the above points, I lean towards negative and suggest the authors to dive deeper according to the revised motivation and revised contributions.

---

### Official Review · Reviewer_6bHT · 2021-11-02

**Correctness:** 4
**Technical Novelty And Significance:** 3
**Empirical Novelty And Significance:** 2
**Recommendation:** 6
**Confidence:** 4

**Main Review:**

Positives
The creation of a simulated flood dataset
The use of segmentation "Masker" conditioned on the depth of an image is something I haven't seen before

The qualitative results appear improved over other methods, and there is a positive human evaluation also

Negatives

the system is a collection of black boxes, depth estimation, segmentation estimation etc

Limited to a single depth of flood value (although this is discussed in future work)

**Summary Of The Paper:**

This topical paper claims to raise awareness of climate change by projecting flooding images of popular places. To achieve this, they use a GAN based image generator placing water on street view style images.

**Summary Of The Review:**

This paper is different  and topical, with still has a contribution to the research community and hopefully will provoke discussions

---

> ### Author Response · Authors · 2021-11-17
> **First reply to reviewer 6bHT**
>
> We are encouraged the reviewer finds our contribution meaningful, original and a proper research contribution. In the following comment, we intend to reply to their review in hope to make our ideas and methods clearer.
>
> In addition, regarding their "novelty" scores, we would like to point them to the global comment we posted to all 3 reviewers, along with our in-depth explanation of the significant novelty our research brings, commented as a reply to other reviewers' concerns on our "two-stage design".
>
> ---
> ### Q1
> > the system is a collection of black boxes, depth estimation, segmentation estimation etc
>
>
> We agree with the Reviewer that explainability is a topic of paramount importance in the development of AI models. However, we find that given that our models are not involved in algorithmic decision-making, using black box approaches is acceptable. Our paper presents a public-facing, applied research project for which explainability is not, in our opinion, so crucial that "black-box" models should be avoided. In addition, we hope our ablation study, while not explaining predictions *per se*, does shed light on the contribution of the individual parts in the model.
>
> ---
> ### Q2
> > Limited to a single depth of flood value (although this is discussed in future work)
>
> As the reviewer notices, we are aware the single water-level is a limitation of our work and there is room for further research in this direction. We agree with them it is a good next step and have already started investigating how to condition the latent representation $z = E(x)$ on a particular expected water-level. This work is currently in progress.

---

> ### Author Response · Authors · 2021-11-19
> **Following up rebuttal**
>
> Dear reviewer,
>
> We know the timeline for the discussion period is tight and we wrote rather lengthy responses to address all aspects of your review. We would love to use this chance to discuss with you any concerns that may remain open. We would appreciate it if you could confirm that you have read our responses and let us know whether we have successfully address the concerns or there are further points to discuss.
>
> Thanks for your time invested in this review!

---

### Author Response · Authors · 2021-11-17
**To all ICLR reviewers: a summary of contributions**

We are sorry we could not make our message clearer both in the motivation and the highlight of our contributions. We hope that with the detailed responses we have provided, reviewers will agree with us that:

* Climate change is an immense risk to modern societies.
* _One_ of the limiting factors of drastic mitigation/adaptation actions is psychological distancing.
* A known and demonstrated reduction factor of this cognitive bias is imagery, even more so personalized visuals.
* AI is required to produce physically plausible and visually realistic images that are customized per location in this context.
* Standard, existing methods are either inapplicable or insufficient on their own.
* This is because, amongst other reasons, this is a brand new task without prior work or even ground truth data.
* We managed to not only successfully re-use existing techniques from adjacent tasks, but also combine, repurpose and improve them to fit our goals.
* To do so we contribute a new, labeled and diverse data set from a virtual world which we used to address the problem.
* And finally that our evaluation procedure is precise, thorough, statistically significant and a contribution in itself absent any other comparable attempts.

We would also hope that reviewers agree with us that research in AI is not just about new blocks or training procedures. It is also about making things work outside their design intentions, it is also about new tasks, new data sets, new applications. We believe the frontiers of AI research and science can also expand towards socially beneficial uses of AI, not _only_ towards ever-improving SOTA metrics or convoluted operations. Our research is certainly atypical, yet our scientific process is rigorous and our contributions meaningful.

---

### Decision · Program_Chairs · 2022-01-20

**Decision:**

Accept (Poster)

**Comment:**

This paper aims at raising awareness of climate change by GAN-projecting flooding images of popular places. This is an interesting case. While all reviews agree that this is an interesting direction, they also value the contributions differently. Two of them would like to see more methodological contributions, two focus more on the contribution to the (psychological) fight of climate change. Nevertheless, the rolling discussion helped to clarify several of the issues raised by the reviewers, and (in my opinion) "combining" existing methods to realize an important model that is one little step towards making people more aware how climate change may impact their own lifes is highly creative and useful. I therefor overall suggest strongly to accept the paper. As the ICLR CfP reads, ICLR is not just about methodological contributions. Societal considerations of representation learning are explicitly mentioned.